# A comparative study of anthropogenic CH$_4$ emissions over China based on the ensembles of bottom-up inventories

Xiaohui Lin[1*], Wen Zhang[1*], Monica Crippa[2], Shushi Peng[3], Pengfei Han[4], Ning Zeng[5], Lijun Yu[1], Guocheng Wang[1]

[1]State Key Laboratory of Atmospheric Boundary Layer Physics and Atmospheric Chemistry, Institute of Atmospheric Physics, Chinese Academy of Sciences, Beijing, China

[2]European Commission, Joint Research Centre (JRC), Ispra, Italy

[3]Sino-French Institute for Earth System Science, College of Urban and Environmental Sciences, Peking University, Beijing, China

[4]State Key Laboratory of Numerical Modeling for Atmospheric Sciences and Geophysical Fluid Dynamics, Institute of Atmospheric Physics, Chinese Academy of Sciences, Beijing, China

[5]Department of Atmospheric and Oceanic Science, and Earth System Science Interdisciplinary Center, University of Maryland, College Park, Maryland, USA

[*] *Correspondence to*: linxh@mail.iap.ac.cn; zhw@mail.iap.ac.cn.

**Abstract**

Atmospheric methane ($CH_4$) is a potent greenhouse gas that is strongly influenced by several human activities. China, as one of the major agricultural and energy production countries, contributes considerably to the global anthropogenic $CH_4$ emissions by rice cultivation, ruminant feeding and coal production,. Understanding the characteristics of China's $CH_4$ emissions is necessary for interpreting source contributions and for further climate change mitigation. However, the scarcity of data from some sources or years and spatially explicit information pose great challenges to completing an analysis of $CH_4$ emissions. This study provides a comprehensive comparison of China's anthropogenic $CH_4$ emissions by synthesizing the most current and publicly available datasets (13 inventories). The results show that anthropogenic $CH_4$ emissions differ widely among inventories, with values ranging from 44.4-57.5 Tg $CH_4$ $yr^{-1}$ in 2010. The discrepancy primarily resulted from the energy sector (27.3-60.0% of total emissions), followed by the agricultural (26.9-50.8%), and waste treatment (8.1-21.2%) sectors. Temporally, emissions among inventories stabilized in the 1990s, but increased significantly thereafter, with annual average growth rates (AAGRs) of 2.6-4.0% during 2000-2010, but slower AAGRs of 0.5-2.2% during 2011-2015, and the emissions became relatively stable with AAGRs of 0.3-0.8% during 2015-2019 because of the stable emissions from energy sector (mainly coal production). Spatially, there are large differences in emissions hotspot identification among inventories, and incomplete information on emission patterns may mislead or bias mitigation efforts for $CH_4$ emission reductions. The availability of detailed activity data for sectors or subsectors and the use of region-specific emission factors play important roles in understanding source contributions, and reducing the uncertainty of bottom-up inventories.

**Keywords:** Anthropogenic $CH_4$ emissions; bottom-up inventories; uncertainty analysis; source and contribution.

**1 Introduction**

Atmospheric methane ($CH_4$) is a potent greenhouse gas with a warming potential that is 28 fold higher than that of $CO_2$ over a 100-year time horizon (Myhre et al., 2013). The global average dry air mole fraction of atmospheric $CH_4$ was 1873.7 parts per billion by volume (ppb) in February 2020 based on marine surface sites (Liu et al., 2015) . $CH_4$ has a relatively short atmospheric lifetime of ~10 years, and reducing $CH_4$ emissions is considered an efficient option to lower radiative forcing in the short term (Montzka et al., 2011; Shindell et al., 2012). The global $CH_4$ budget is strongly influenced by several human activities, including food production (ruminant and rice), waste (sewage and landfills), and fossil fuel production and use (coal, oil and gas) (Bruhwiler et al., 2014; Menon et al., 2007). Global anthropogenic $CH_4$ emissions (~357 Tg $CH_4$ $yr^{-1}$) contributed approximately 60% of the total emissions, as estimated by atmospheric inversions (Saunois et al., 2020). According to the latest report from a global methane project, emissions from agriculture contributed the most (44%) to global anthropogenic sources, followed by fossil fuel (35%) and waste (12%) (Saunois et al., 2020). Control of anthropogenic $CH_4$ emissions has become a promising target in the effort to mitigate climate change at short timescales

(Höglund-Isaksson, 2012; Henne et al., 2016; Saunois et al., 2016). Therefore, understanding the levels and trends of anthropogenic $CH_4$ emissions and their drivers is extremely crucial for global change research and mitigation.

The estimation of anthropogenic $CH_4$ emissions is extremely challenging, due to the complexity of the processes included and difficult to quantify separately (Saunois et al., 2020). Considerable uncertainties are caused by source-specific information combined with activity data and emission factors (Henne et al., 2016; Zhang et al., 2018). Using coal mining as an example, the time dynamic information of geolocation, emission factors and production of coal mines are rather insufficient for $CH_4$ emission quantification (Sheng et al., 2019).The current estimates of global anthropogenic emissions

ranged from 334 to 375 Tg $CH_4$ $yr^{-1}$ by top-down approaches and from 348 to 392 Tg $CH_4$ $yr^{-1}$ by bottom-up approaches during 2008-2017 (Saunois et al., 2020). Top-down (atmospheric inversions) approaches provide a good picture of global and continental $CH_4$ emissions (Alexe et al., 2014). However, for small-scale regions, inversions largely depend on prior emission inventories and are still limited by their coarse spatial resolutions (Alexe et al., 2014; Henne et al., 2016). To improve the spatial resolution and representation of top-down inversions, more efforts have been made at regional scales

(Thompson et al., 2015; Wecht et al., 2014), but it is still difficult to mechanistically model $CH_4$ emissions from a particular type of emissions source (Cui et al., 2015; Kirschke et al., 2013). Bottom-up emissions estimates are based on source-specific information on activity data and emission factors. The analyses of source-specific emissions help us understand the relationship between emissions and the underlying socioeconomic and sociodemographic driving processes (Miller and Michalak, 2017; Zhou and Gurney, 2011). Bottom-up inventories are essential in terms of providing baseline

information on emission characteristics, and reliable emission estimates can further help with optimizing mitigation strategies (Cheng et al., 2014; Sheng et al., 2019). However, the accuracy of bottom-up inventories largely depends on the reliability of activity data and emission factors. Global inventories are generally based on country-level activity data and emission factors, which hardly fully characterize the regional discrepancies caused by the large variability of socioeconomic characteristics (Bergamaschi et al., 2010; Peng et al., 2016; Zhu et al., 2017).

As a country with widespread rice and coal production areas and a growing human population with billions of people, China is a large emitter of $CH_4$ (Ito et al., 2019; Janssens-Maenhout et al., 2019). The main anthropogenic sources of $CH_4$ in China in 2014, as reported by the National Communication on Climate Change (NCCC) of the People's Republic of China, were energy (45% of anthropogenic emissions), agriculture (40%), and waste (12%) . However, anthropogenic $CH_4$ emissions differ widely among inventories with differences as high as 17 Tg $CH_4$ found for 2010 (Ito et al., 2019), of which paddy and

coal mining emissions contributed a large part of the differences (Cheewaphongphan et al., 2019). Due to the scarcity of data from some sources or years and spatially explicit information, a quantitative analysis of China's $CH_4$ emissions remains a great challenge. Several studies have quantified the emissions from rice paddies in China by using process-based modeling approaches (Huang et al., 1998; Li et al., 2002; Tian et al., 2011; Zhang et al., 2011). However, there are considerable

differences in the modeling estimates. In CH4MOD model, the estimated $CH_4$ emissions from rice paddies varied from 3.8 to 9.8 Tg, of which 56.6% resulted from model fallacy, and the remaining 43.4% was attributed to errors and the scarcity of input data (Zhang et al., 2017). As the largest coal producer worldwide, China's coal mine $CH_4$ emissions are still poorly quantified, and estimates vary significantly from 14 to 28 Tg $CH_4$ $yr^{-1}$ (Sheng et al., 2019). In addition, emissions from waste treatment are mainly focused on the total emissions of city-level or provincial wastewater in China (Du et al., 2018; Zhao et al., 2019). Emissions from Chinese landfills are estimated by Cai et al. (2018) and Du et al. (2017), but there remain gaps in spatial or temporal coverage. Altogether, there have been few studies on the comprehensive evaluation of China's anthropogenic $CH_4$ emissions, although one or several representative emission sources have been studied at the provincial level or in certain regions (Chen et al., 2011; Huang et al., 2019; Liu et al., 2016; Ren et al., 2011; Yue et al., 2012; Zhang and Chen, 2014). Therefore, comprehensive analysis by gathering existing inventories is particularly important to improve the understanding of China's contribution to the global $CH_4$ budget and to provide guidance on mitigation policies.

Based on a comprehensive literature review of previous studies, we have included the most current and publicly available datasets (13 global and regional inventories) to characterize anthropogenic $CH_4$ emissions in China. We presented a detailed evaluation of the major emission sectors, including agricultural activities (rice cultivation and livestock), energy activities (fossil fuel production and use), and waste management (wastewater and landfill), in the existing inventories (Table 1). The specific objectives of this study were to (1) adequately understand the characteristics and dynamics of anthropogenic $CH_4$ emissions in China and identify their sectoral and regional contributions and (2) understand sources of discrepancies among inventories and provide helpful suggestions for further improvements in estimations and policy-making related to the control of $CH_4$ emissions.

## 2 Data and Methods

Here, we collected 13 global and regional bottom-up inventories for anthropogenic $CH_4$ emissions over mainland China (listed in Table 1), including 5 gridded datasets and 8 statistical datasets. Specifically, the 5 gridded inventories were collected from Peking University (PKU-CH4-China-v1) (Peng et al., 2016), Community Emission Data System (CEDS v2017-5-18) developed for use by the climate modeling community in the Coupled Model Intercomparison Project Phase 6 (CMIP6) (Hoesly et al., 2018), Emissions Database for Global Atmospheric Research (EDGAR v5.0) developed by the European Commission's Joint Research Centre (JRC) and the Netherlands Environmental Assessment Agency (PBL) (Crippa et al., 2019), Greenhouse Gas and Air Pollution Interactions and Synergies (GAINS/ECLIPSE v5a CLE baseline) developed by the International Institute for Applied Systems Analysis (IIASA) (Höglund-Isaksson, 2012), and Regional Emission Inventory in ASia (REAS 2.1) (Kurokawa et al., 2013; Ohara et al., 2007). The latest version of CEDSv2021-02-05 (only tabular data) was also included to understand the emissions trend in recent years through personal communications.

PKU is a global annual bottom-up inventory of anthropogenic $CH_4$ emissions from 1980 to 2010 that compiles regional sector-specific emission factors with provincial emissions from the eight major source sectors in China (Peng et al., 2016). CEDS implements a mosaic approach to produce monthly country emissions from 16 sectors and 53 subsectors based on existing emission inventories, emission factors, and activity data (e.g., EDGAR v4.2, GAINS) during the period of 1970-2014 (Hoesly et al., 2018). EDGAR v5.0 provides annual country emissions through 24 sectors specified by the Intergovernmental Panel on Climate Change (IPCC) from 1970 to 2015. The GAINS model identifies forty source sectors for $CH_4$ and estimates region-specific emissions for the period of 1990-2010 at five-year intervals, with projections to 2030 (Höglund-Isaksson, 2012). REAS provides a monthly Asian inventory of anthropogenic emission sources from 14 sectors for $CH_4$ from 2000 to 2008 (Kurokawa et al., 2013). The 8 statistical tabular data sets used in this study were from research institutes and published literature, including the Environmental Protection Agency (EPA) of the United States; Food and Agriculture Organization (FAO); NCCC of the People's Republic of China; Global Methane Budget (GMB) released by the Global Carbon Project (Saunois et al., 2020), and GMB has a bit overlap with the other datasets used here, but to keep the completeness of this important work, we kept all the inventories to produce the GMB estimates; published literature data from Yue et al. (2012), Huang et al. (2019), Zhang and Chen (2014), Zhang et al. (2016), Zhang et al. (2018), and China High Resolution Emission Database (CHRED) (Cai et al., 2018). To analyze the spatiotemporal patterns and discrepancies among inventories, specific anthropogenic sectors were aggregated into 3 categories (i.e., agriculture, energy, and waste) (Table S2).

Generally, bottom-up inventories are based on national or subnational level activity data and emission factors. The four gridded emissions (i.e., CEDS, EDGARv5.0, GAINS, and REAS) are generally based on country-specific socioeconomic statistics and with country-level or Intergovernmental Panel on Climate Change (IPCC) default emission factors (Crippa et al., 2019; Höglund-Isaksson, 2012; Kurokawa et al., 2013; Ohara et al., 2007), which are widely used as priori emissions for atmospheric research. The PKU inventories for China considered regional discrepancies by applying province-level (Fig. S1) activity data from the National Bureau of Statistics of China (NBS) and region-specific emission factors when data availability allowed, especially for provinces with large differences in economic development (Peng et al., 2016). To quantify the spatial consistency among inventories, the kappa coefficient is used to analyze the degree of agreement between two estimates. Here, PKU was used as a reference to check the consistency with the remaining inventories. A kappa of value equal to 1 indicates perfect agreement, whereas a value of 0 indicates no agreement beyond chance (Landis and Koch, 1977). Spatially, high emissions areas are critical for targeting $CH_4$ emission reductions, and the top 2% high-emitting grids (> 33 g $CH_4$ m$^{-2}$ yr$^{-1}$) from PKU are considered as emissions hotspots to assess the capability of emissions hotspot identification among inventories. Further details of the tabular datasets used in this study are listed in Table S1. Detailed information on sector and subsector categories for inventories is provided in Table S2. To improve the understanding of the recent trends in

China's CH$_4$ emissions, we estimated emissions from 2015-2019 using IPCC Tier 1 method based on national activity data from NBS (NBS, 2021) and localized optimized emission factors from NCCC (Table S5-S7).

Table 1 Key features of gridded emissions inventories

| Name (version) | PKU (PKU-CH4-China-v1) | CEDS (CEDS v2017-05-18) | EDGAR (EDGARv5.0) | GAINS (ECLIPSE V5a) | REAS (REAS 2.1) |
|---|---|---|---|---|---|
| Year | 1980-2010 | 1970-2014 | 1970-2015 | 1990-2050 at 5-year intervals | 2000-2008 |
| Domain | Global | Global | Global | Global | East, Southeast, South, and Central Asia |
| Spatial resolution | 0.1 | 0.5 | 0.1 | 0.5 | 0.25 |
| Temporal resolution | Annual | Monthly | Annual | Annual | Monthly |
| Sources of activity data | | | | | |
| Agriculture | Provincial agriculture statistics (National Bureau of Statistics of China, NBS) | EDGAR v4.2 | FAO | FAO | FAO |
| Energy | Provincial energy statistics (NBS) | IEA; EDGAR v4.2; ECLIPSE v5a | IEA | IEA | IEA, Provincial energy statistics (NBS) |
| Waste | Provincial environmental statistics (NBS) | FAO; EDGAR v4.2 | UNFCCC | UNFCCC,FAO | NA |
| Data access | http://inventory.pku.edu.cn/home.html | http://www.globalchange.umd.edu/ceds/ceds-cmip6-data/ | https://edgar.jrc.ec.europa.eu/overview.php?v=50_GHG | https://iiasa.ac.at/web/home/research/researchPrograms/air/ECLIPSEv5a.html | http://www.nies.go.jp/REAS/index.html#data%20sets |
| Reference | Peng et al. (2016) | Hoesly et al. (2018) | Crippa et al.(2019) | Höglund-Isaksson (2012) | Kurokawa et al. (2013) |

*The complete list of data sources can be found in the References.

**3 Results and discussions**

3.1 Temporal variations of anthropogenic CH$_4$ emissions

The anthropogenic CH$_4$ emissions from China differ widely among inventories, and emission estimates are in the ranges of 28.5-46.3 and 44.4-57.6Tg CH$_4$ yr$^{-1}$ for 1990 and 2010, respectively, but are still broadly within the minimum-maximum range of the GMB for 2000-2009 and 2003-2012 (Fig. 1). The existing inventories show rather consistent temporal trends. CH$_4$ emissions stabilized in the 1990s but increased significantly thereafter, with AAGRs of 2.6% (EDGAR) – 4.0%

(CEDSv2021-02-05) during 2000-2010, and slower AAGRs of 0.5% (EDGAR) - 2.2% (FAO) during 2011-2015. The estimated emissions in this study using national-level activity data from the NBS and localized emission factors from NCCC increased slowly from 50.7 Tg $CH_4$ $yr^{-1}$ to 52.3 Tg $CH_4$ $yr^{-1}$ (AAGRs = 0.8%) during 2015-2019. This estimate showed a slightly increasing trend of 0.5 Tg $CH_4$ $yr^{-2}$ for the period of 2015-2019, which is rather consistent with the values of 0.3±0.1 Tg $CH_4$ $yr^{-2}$) from the top-down approach by (Sheng et al., 2020) and 0.3 Tg $CH_4$ $yr^{-2}$ from CEDSv2021-02-05. The coal sector appears to be a major driver of the trend in China's $CH_4$ emissions, and a clear increasing trend (1.0±0.3 Tg $CH_4$ $yr^{-2}$) was found during 2012-2015 (Miller et al., 2019). The emissions from coal production showed a slight increasing trend (0.3 Tg $CH_4$ $yr^{-2}$), with AAGRs of 1.0% during 2015-2019 in this study. A small growth in coal mine emissions was also found in the study of Sheng et al., (2020) and CEDSv2021-02-05. Specifically, during 2000-2010, emissions from the existing inventories increased from 37.2±5.8 Tg $CH_4$ $yr^{-1}$ to 49.6±4.5 Tg $CH_4$ $yr^{-1}$. The growth of $CH_4$ emissions is attributed mostly to an increase in emissions from the energy sector, with AAGRs of 5.8% - 9.0% (Fig. S2). A considerable discrepancy was found between REAS and the other inventories in terms of the magnitude and variation, with a difference as high as 35.8 Tg $CH_4$ in 2008. Furthermore, emissions from the energy sector in REAS were ~2 times greater than those from other inventories (22-24 Tg $CH_4$ $yr^{-1}$). The trend in REAS was mostly triggered by a fast increase in energy sector emissions, with AAGRs greater than 10% during 2000-2008. This result was probably because the coal consumption trend was adjusted to a higher value in the China Statistical Yearbook (CSY), according to the GOME satellite, with a higher trend (increased 50% from 1996-2002) than the provincial statistical trend (25%) and IEA trend (15%) (Akimoto et al., 2006; Ohara et al., 2007). The $CH_4$ emissions estimated from EDGAR v5.0 were 13.2% higher than those from NCCC, in the respective corresponding periods. These results are due to the higher estimates of agriculture and energy emissions obtained by using higher emission factors in rice cultivation and coal mining in EDGAR (Cheewaphongphan et al., 2019; Peng et al., 2016). For coal mining, the emission factor used in EDGAR is 10.0 $m^3$ $t^{-1}$, while NCCC is a lower 8.89 $m^3$ $t^{-1}$, and for rice cultivation, EDGAR is 0.1-1.4 g $m^{-2}$ $d^{-1}$, while NCCC is 0.005-0.21 g $m^{-2}$ $d^{-1}$ (Table S4). Emissions derived from PKU were 12.2% lower than those from NCCC, which resulted from the lower emission factors in livestock and coal mining (NDRC, 2014; Peng et al., 2016). Therefore, the provincial emission factors in Table S6 for coal mining emissions are useful in the improvement of national-data-based inventories.

Specifically, agricultural activities were the main contributors to national $CH_4$ emissions before 2000 (46.1-60.0% of the total emissions, Fig. S2), as reported by the FAO. Emissions from agriculture were rather stable and showed slight decreases during 2000-2010, with AAGRs of -0.7~-0.5% among the inventories. This result is caused by the decreasing trend of emissions from rice production and livestock, with AAGRs of -0.03~-0.8% and -0.5~-0.7%, respectively. However, EDGAR v5.0 and CEDSv2021-02-05 presented an increasing trend in agriculture (AAGR = 0.2% and 1.5%) in the same period, which resulted from the combined effect of emissions growth in rice production (AAGR=0.9%), a reduction in livestock

(AAGR=-0.6%) in EDGAR v5.0, and a dominating increasing trend in livestock in CEDSv2021-02-05 (AAGR=2.3%) (Fig. S3). Over the study period, energy source emissions showed a substantial increase, ranging from 11.0±3.0 Tg CH$_4$ yr$^{-1}$ in 1990 to 24.0±2.4 Tg CH$_4$ yr$^{-1}$ in 2010. After 2000, emissions from energy increased significantly and became the leading source (AAGR: 5.9-9.0%, 2000-2010). This increase was mainly driven by the rapid growth of coal production in China, with an AAGR up to 9.0% in the 2000s, while it was only 2.6% in the 1990s according to the official data released by the National Bureau of Statistics of China (CSY, 2019). However, China has consolidated its coal industry to concentrate production by transforming small mines into larger and more efficient coal mines (abandoning approximately 4000 mines) since 2010 (Sheng et al., 2019; Sheng et al., 2020). As a result, the emissions from coal mines have stabilized or decreased since 2012, with coal production in 2016 returning to levels similar to those in 2010 (~2.4×10$^3$ million tons) (CSY, 2019; Sheng et al., 2020). Additionally, discrepancies exist in the magnitude of waste sector emissions, although the value continued to increase steadily during 2000-2010 (AAGR: 2.1-3.4%).

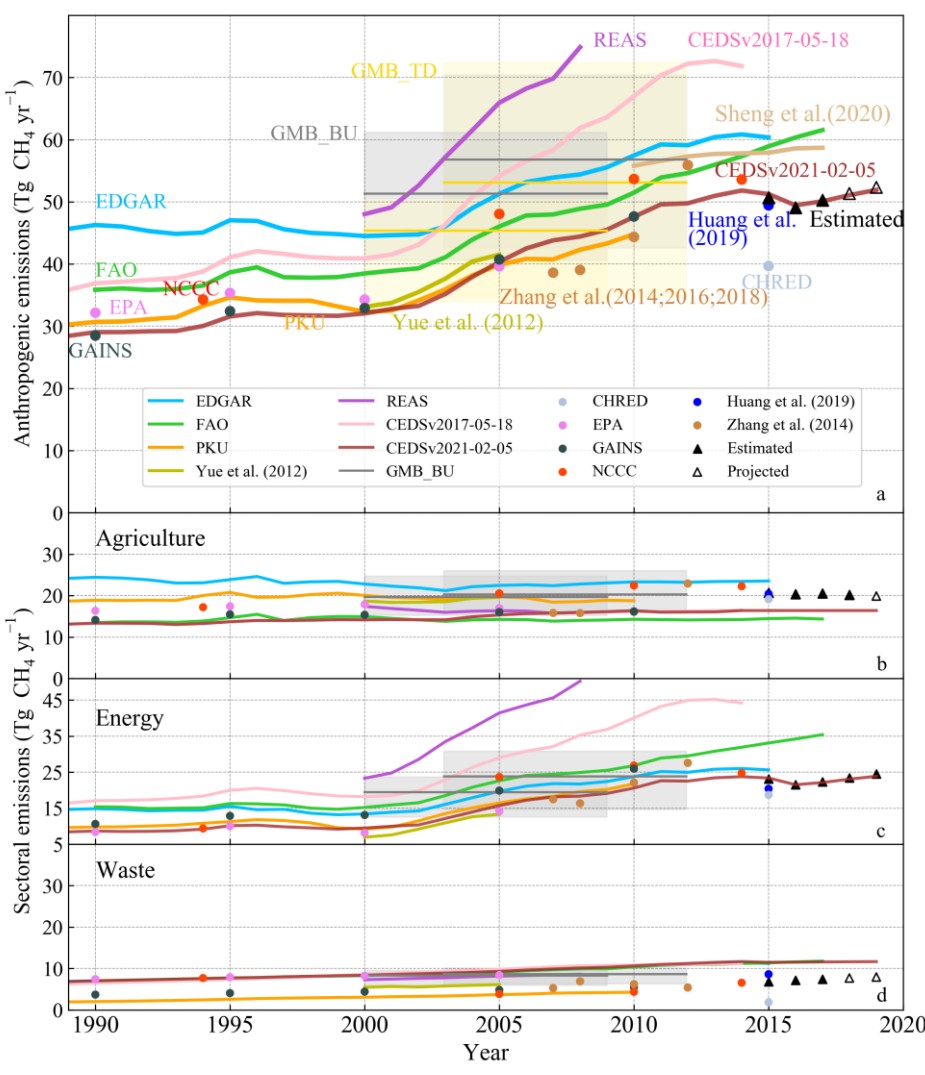

Fig. 1 The temporal variation in China's total (a) and sector-specific (c-d) CH$_4$ emissions since 1990. Gray and yellow lines

indicate the mean of the bottom-up and top-down estimates of $CH_4$ emissions from GMB, respectively. Shaded areas represent the min-max value of emissions from GMB. The emissions from 2015 to 2019 in this study (black triangles) refer to estimates using national activity data from the NBS and localized emission factors from NCCC. Note that the empty triangle indicates projected values using the trend over the last 5 years.

3.2 Spatial patterns of anthropogenic $CH_4$ emissions

Available gridded emissions remain limited; thus, the spatial pattern analysis of $CH_4$ emissions was performed on the PKU, CEDSv2017-05-18, EDGAR v5.0, GAINS, and REAS inventories (Fig. 2, Table 1). In 2010, China's $CH_4$ emissions were dominated by emissions from the energy sector (41-67% of total emissions), followed by emissions from agricultural activities (21-42%), and waste treatment (10-18%) (Fig. S2). To interpret the discrepancy in emissions among different inventories, frequency distribution and kappa analysis were conducted at the grid cell level (Fig. 3). The higher kappa coefficient of 0.51 indicates that EDGAR has a relatively better agreement with PKU than those from CEDS and GAINS (0.43 and 0.40). REAS had a weak correlation with PKU, with a kappa coefficient of 0.30. Remarkable regional disparities were observed among inventories. The spatial patterns had a close relationship with regional urbanization and economic activities, because of the associated increased energy production and livestock and waste sector emissions. High-emissions areas (e.g., emitting grids $> 40$ g $CH_4$ m$^{-2}$ yr$^{-1}$) were generally located in densely populated areas (such as Beijing and Shanghai), energy production regions (such as Shanxi), and rice cultivation areas in south-central China as well as livestock-dominated regions in the North China Plain and Northeast China.The western regions showed low emissions (e.g., emitting grids $< 1$ g $CH_4$ m$^{-2}$ yr$^{-1}$). Intense emissions from large cities are attributable to industrial activities, transportation, and solid waste in landfills (Ito et al., 2019). The expansive areas of rice paddy and double-cropping systems in southern and central China are recognized as being large contributions to the corresponding high emissions (Chen et al., 2013; Zhang et al., 2011). Due to massive emissions from coal mining, provinces such as Shanxi, Ningxia, Henan, Guizhou, Chongqing, and Sichuan were emissions hotspots, emitting grids higher than 40 g $CH_4$ m$^{-2}$ yr$^{-1}$. To further characterize the spatial distribution of emissions hotspots, the top 2% high-emitting grids ($> 33$ g $CH_4$ m$^{-2}$ yr$^{-1}$) based on PKU were analyzed to identify the consistency and differences among inventories (Fig. 2I-V). Regional emissions hotspots were presented in PKU and EDGAR (Fig. 2I, III), suggesting the capability of identifying high-emission areas in the North China Plain and southern agricultural areas. However, such patterns showed a large spatial heterogeneity among inventories. There was a lack of emissions hotspots in southern China in GAINS (Fig. 2IV). Specifically, PKU and EDGAR both showed a large number ($>1000$, Fig. 2I, III) of high-emitting grids (emissions $> 33$ g $CH_4$ m$^{-2}$ yr$^{-1}$), accounting for 27% and 41% of total emissions. However, the numbers of high-emitting grids from CEDS and GAINS were only 89 and 48 (Fig. 2II, IV), accounting for 50% and 16% of total emissions, respectively. In addition, the number of high-emitting grids (32% of total emissions) from REAS was less

than half that from PKU and EDGAR (Fig. 2V). This indicated that CEDS and GAINS can not properly interpret hotspots. Emission hotspots in REAS were strongly biased towards Shanxi Province. The incomplete information on emission patterns may mislead or bias mitigation efforts for $CH_4$ emission reductions.

There were substantial discrepancies in the magnitude and distribution of sector-specific emissions among the inventories. For example, the amount of $CH_4$ emissions from agriculture in EDGAR v5.0 was 24.2-45.7% higher than those from PKU, CEDS, REAS, and GAINS. The spatial pattern of agricultural emissions in EDGAR was similar to the corresponding distribution in PKU because the distribution of rice and livestock both used the gridded data from Monfreda et al. (2008) and (Robinson et al., 2007), and further the emission factors of rice cultivation used in EDGAR were updated with those in PKU (Janssens-Maenhout et al., 2019). Grids with high estimations (10-40 g $CH_4$ $m^{-2}$) were mainly located in the Yangtze River valley (Fig. 2i) and the eastern part of the Beijing-Tianjin-Hebei region accounted for nearly half of the agricultural emissions (with values that were 22.7-39.3% higher than the others, Fig. 2v). The higher $CH_4$ emissions estimated from EDGAR v5.0 in Beijing is due to the higher number of livestock from FAO statistics (5.5 million cattle) (Gilbert et al., 2018), which was considerably higher than the number provided by NBS (0.3 million cattle) in 2010 (CSY, 2019). Additionally, GAINS and REAS tended to allocate more emissions from energy to the North China Plain (such as Shanxi and Shandong provinces, Fig. 2n and 2s). More than 75% of the energy emissions from EDGAR v5.0 were allocated in high-emitting grids (>60 g $CH_4$ $m^{-2}$ $yr^{-1}$, Fig. 2w), which covered less than 0.8% of the total number of grids. This result implied that EDGAR may provide lower estimates in other areas. EDGAR v4.2 originally uses 328 coal mines with locations for China from World Coal Association as point emissions to disaggregate the amount of national emissions (Janssens-Maenhout et al., 2013) and is then updated using data from Liu et al. (2015). However, emissions from coal mining estimated by EDGAR v5.0 still have notable bias towards Shanxi Province (Fig. 5f). Emissions from the energy sector in CEDS have a similar pattern as EDGAR, with 72% energy emissions from high-emitting grids (>60 g $CH_4$ $m^{-2}$ $yr^{-1}$, Fig. 2f,w). The data source of CEDS is mainly from EDGAR v4.2 (Hoesly et al., 2018). PKU had a distinct spatial pattern for energy emissions (Fig. 2b), which was attributable to the fact that emissions from coal exploitation were located using the geolocation (latitude and longitude) of 4264 coal mines from Liu et al. (2015) and the regional emission factors (Peng et al., 2016). Emissions from waste treatment were mostly located in more developed areas, such as the North China Plain, Yangtze River Delta and Pearl River Delta. Zhang and Chen (2014) also found that emissions from waste treatment were related to the size of the economies of the regions and their urban population scales to a certain extent. The emissions from waste treatment estimated by EDGAR v5.0 and CEDS were 20.7-152.5% higher than the values from other inventories. Moreover, EDGAR v5.0 tended to have higher emissions from waste treatment in urban areas, whose emission hotspots (> 33 g $CH_4$ $m^{-2}$ $yr^{-1}$) were highly consistent with the distribution of provincial capitals (Fig. 2k,III). Higher emissions of waste treatment in EDGAR were from wastewater, which probably adopted a higher $CH_4$ correction factor for wastewater treatment plants or a higher chemical oxygen demand

(Peng et al., 2016).

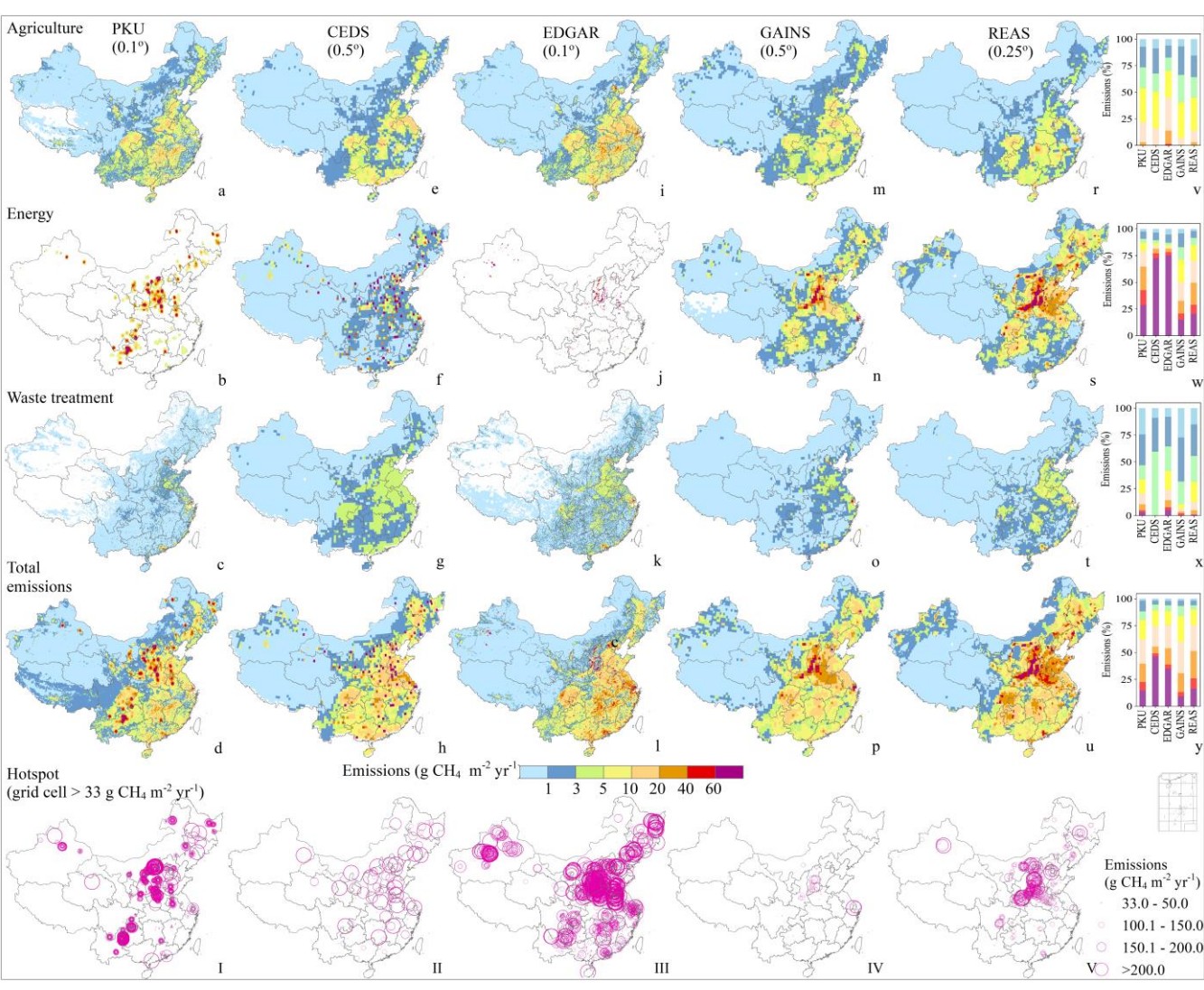

Fig. 2 The spatial distribution of sectoral and total anthropogenic CH$_4$ emissions from PKU (a-d), CEDSv2017-05-18 (e-h), EDGAR v5.0 (i-l), GAINS (m-p) in 2010 and REAS (q-t) in 2008, and emissions frequency (u-x). The top 2% high-emitting grids (emissions > 33 g CH$_4$ m$^{-2}$ yr$^{-1}$) were based on PKU.

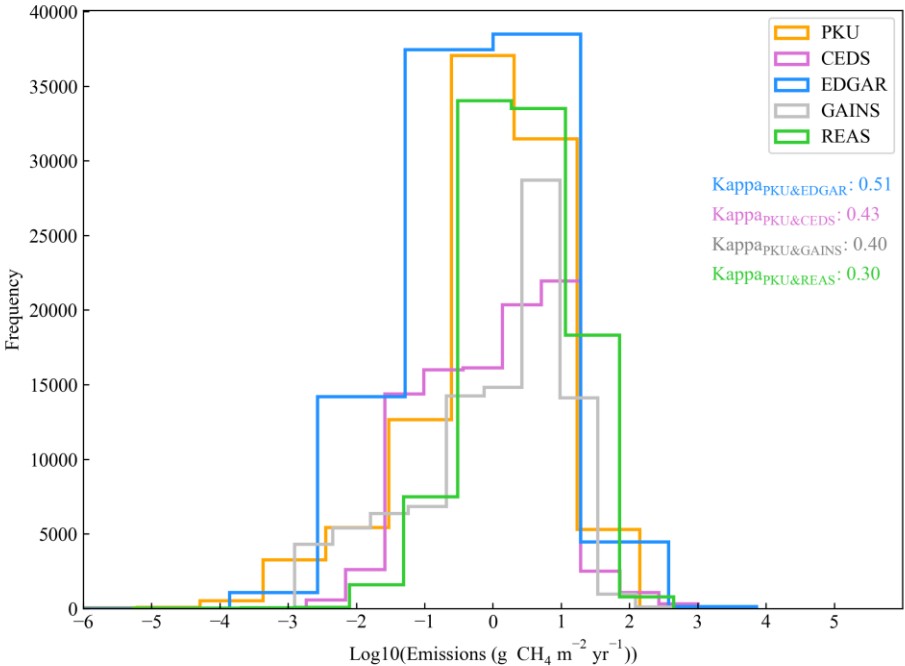

Fig. 3 Frequency counts of emitting grids for PKU, CEDSv2017-05-18, EDGARv5.0, and GAINS in 2010, and REAS in

2008. Kappa coefficients were calculated based on the quartile of PKU.

3.3 Changes in the spatial pattern of anthropogenic $CH_4$ emissions from 2000 to 2010

From 2000 to 2010, anthropogenic $CH_4$ emissions increased considerably in China, and this increase was mainly driven by increased emissions from energy exploitation (especially in coal mining) in the northern and central regions, followed by waste treatment in the southern and eastern regions and agriculture in the northeastern region (Fig. 4). Growth was

275 profoundly affected by urbanization and economic development. The decrease in $CH_4$ emissions from PKU in southern and southeastern China was attributed to a decline in rice cultivation and livestock feeding (Peng et al., 2016), and similar results were also observed in REAS (Fig. 4a,q). Since the 1980s and perhaps earlier, most Chinese farmers have adopted the practice of draining paddy fields in the middle of the rice-growing season, which halts most of the methane released from the fields (Qiu, 2009). Additionally, emissions from livestock in southeastern China have decreased due to the reduction in the

280 buffalo population (Yu et al., 2018). These changes in livestock and rice cultivation contributed to mitigating in $CH_4$ emissions. In EDGAR v5.0, a decreasing trend was found for energy emissions in the central regions and in the North China Plain (Fig. 3j), while a similar trend was not found in the other inventories during 2000-2010. These results were attributed to the reduced emissions in the subsector of energy for buildings (RCO, Fig. S4). In addition, Shanxi Province had a larger contribution to the changes in energy emissions in EDGAR v5.0 (40%) than to those in other inventories (18-23%), which

285 may have omitted emissions in other regions.

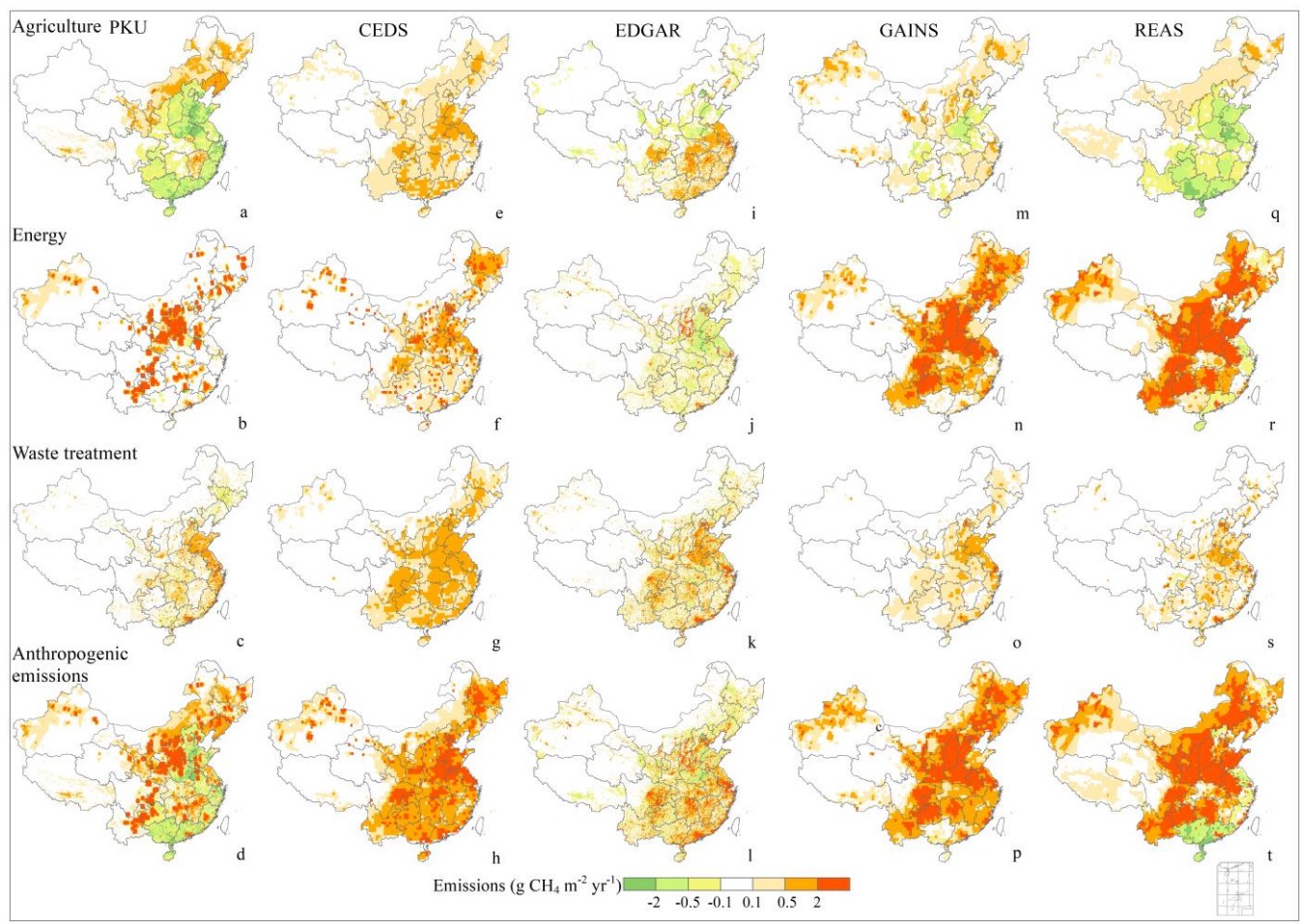

Fig. 4 Changes in sectoral and total anthropogenic CH$_4$ emissions from PKU (a-d), CEDSv2017-05-18 (e-h), EDGAR v5.0 (i-l), GAINS (m-p) from 2000 to 2010, and REAS (q-t) from 2000 to 2008.

290 3.4 Further comparison with other inventories at the subsector level

To further evaluate the quality of existing inventories, independent and more detailed subsector datasets were collected to improve our understanding of the uncertainty in total amounts and spatial patterns among different inventories. Based on the data availability, three subsectors of major emissions sources are displayed, i.e., rice cultivation, livestock, and coal mining (Fig. 5). These three subsectors accounted for 70-85% of the total emissions in China in 2010. The data used for comparison 295 were collected from Zhang et al. (2017) (for rice cultivation), Lin et al. (2011) (for livestock), and Sheng et al. (2019) (for coal mining). Zhang et al. (2017) compiled the NCCC inventory of rice by using a semiempirical model (CH4MOD). The CH4MOD model is a semiempirical model simulating CH$_4$ production and emissions at daily steps. Inputs into the CH4MOD include daily air temperature, percentage of sand in the paddy soil, rice grain yield, type and amount of organic matter applied, and water management used for rice irrigation (Zhang et al., 2011). Lin et al. (2011) estimated emissions 300 from livestock based on county-level statistical data and region-specific emission factors. Sheng et al. (2019) estimated

emissions from coal mining based on more than 10000 operating coal mines reported by the Chinese State Administration of Coal Mine Safety (SACMS).

For the rice cultivation subsector, the amount from PKU was 7.3 Tg $CH_4$ $yr^{-1}$, which is comparable to the value of 8.2 Tg $CH_4$ $yr^{-1}$ reported for 2010 by Zhang et al. (2017) (Fig. 5j). However, EDGAR v5.0 tended to provide higher estimates, with a value of 13.9 Tg $CH_4$ $yr^{-1}$ (Fig. 5d). This difference could be seen from the larger contribution of high-emitting grids ( $> 10$ g $CH_4$ $m^{-2}$ $yr^{-1}$, Fig. 4m) in EDGAR v5.0 (6.7 Tg $CH_4$ $yr^{-1}$ or 48.7% of total emissions), while the values in the other inventories ranged from 17~34% (1.2~2.8 Tg $CH_4$ $yr^{-1}$). The higher estimates from EDGAR v5.0 were primarily located in the Yangtze River (e.g., Hunan and Jiangxi). According to the study of Cheewaphongphan et al. (2019), EDGAR uses a higher proportion of continuous floods, leading to a higher emission factor than that produced in intermittent flood conditions. In contrast, REAS tended to provide a lower estimate (6.7 Tg), especially in the Yangtze River and Northeast China (Fig. 5g). This discrepancy is partly because emissions from rice cultivation in REAS2.1 are from 2008, while others are from 2010. Moreover, emissions in 2008 from REAS2.1 are extrapolated from REAS1.1 in 2000 (Kurokawa et al., 2013), which may not have captured the emission changes caused by the increases in rice cultivation area. As reported by the NBS, the areas of rice cultivation have increased by 5900 $km^2$ in Anhui, Hunan, Jiangsu and Jiangxi provinces, and 12,514 $km^2$ in Northeast China (i.e., Heilongjiang, Jilin, and Liaoning provinces) from 2000 to 2008 (CSY, 2019). Overall, PKU and Zhang et al. (2017) were closer to the NCCC estimates with provincial activity data and emission factors, and Zhang et al., (2017) used the detailed regional water management data and provincial organic matter application rates, which is also used in NCCC as part of the national inventory reported to the UNFCCC (NCCC, 2018).

For the livestock subsector, including enteric fermentation and manure management (Chang et al., 2019), the amount of emissions ranged from 9.2 (REAS) to 11.4 (PKU) Tg $CH_4$ $yr^{-1}$. The bottom-up inventory based on detailed county-level activity data estimated the 2010 emissions to be 12.4 Tg $CH_4$ $yr^{-1}$ (Lin et al., 2011). A consistent spatial pattern from livestock sources was found among inventories. However, REAS had lower   emissions in the North China Plain (such as in Shandong and Henan), Tibetan Plateau and Northeast China, which missed large numbers of high-emitting grids compared to other inventories (Fig. 5h). In addition, higher emissions in the northeastern part of Beijing were reported by EDGAR v5.0, with grids emitting more than 20 g $CH_4$ $m^{-2}$ $yr^{-1}$ (Fig. 5e). These results were caused by the high estimated number of livestock induced by using machine learning methods in the spatial proxy approach (Gilbert et al., 2018).

For the coal mining subsector, the amounts from PKU and EDGAR v5.0 were 17.3 and 19.0 Tg $CH_4$ $yr^{-1}$ in 2010, respectively, which were comparable to the values of 16.7 Tg $CH_4$ $yr^{-1}$ in 2011 from Sheng et al. (2019) and 16.0 Tg $CH_4$ $yr^{-1}$ in 2010 from Zhu et al. (2017). However, emissions from REAS showed a large difference from those in the other inventories, with values up to 38.4 Tg $CH_4$ $yr^{-1}$ in 2008. Spatially, more than 92% of emissions from coal mining in EDGAR v5.0 were located in high-emitting grids (>60 g $CH_4$ $m^{-2}$, Fig. 5d), which covered less than 0.5% of the total grid number.

This result may be due to the limited number of coal mines (~ 4000) used in EDGAR (Crippa et al., 2019; Sheng et al., 2019). The allocation of national total emissions to limited mine locations leads to incorrect spatial patterns and artificial emission hot spots (Sheng et al., 2019). These spatial errors would cause bias in the analysis of trends and source attribution

in inversions, and mislead mitigation strategies in coal exploitation (Sheng et al., 2019). Additionally, emissions from coal mining in PKU show a relatively consistent pattern with that in Sheng et al., (2019); however, PKU tended to have similar proportions among emitting grids (Fig. 5o). This result is because the locations of coal mines used in PKU have a coarser spatial resolution than 0.1˚.

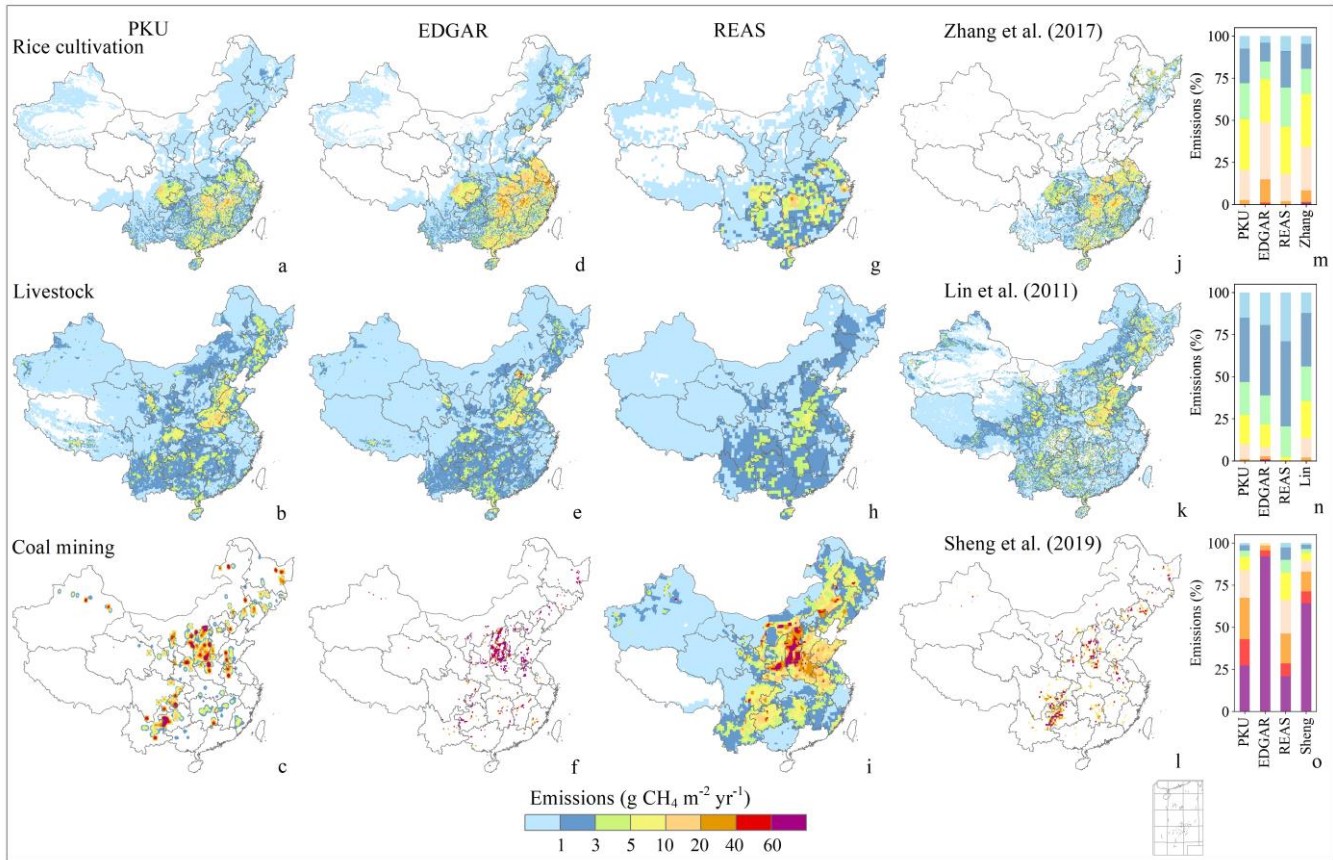

Fig. 5 The spatial distribution of subsectoral $CH_4$ emissions among inventories in 2010. Emissions from coal mining in EDGAR v5.0 were aggregated to a spatial resolution of 0.2˚.

3.5 Estimates and uncertainties of total and sectoral emissions

Considering the comparability of different inventories (i.e., with the same year (2010), and completeness of all same

subsectors), emissions were collected for five datasets (i.e., PKU, EDGAR v5.0, CEDSv2021-02-05, NCCC, and Zhang et al. (2016)). In 2010, the total emissions in China were estimated to be 49.6±4.5 Tg $CH_4$ $yr^{-1}$ (mean ± standard deviation (SD), hereafter the same) among inventories (Fig. 6a). The mean emissions from agricultural activities were 18.5±3.1 Tg $CH_4$ $yr^{-1}$, of which livestock contributed 11.0 Tg $CH_4$ $yr^{-1}$ and rice cultivation contributed 7.8 Tg $CH_4$ $yr^{-1}$ (Table S3). Among all the

agricultural activities, rice cultivation showed a relatively large range from 5.3 Tg $CH_4$ $yr^{-1}$ in CEDSv2021-02-05 to 13.9 Tg $CH_4$ $yr^{-1}$ in EDGAR v5.0 (Fig. 6b). The $CH_4$ emissions from rice paddies are among the most uncertain estimates in rice-growing countries (Huang et al., 2006). High spatial heterogeneity and inadequate data on rice cultivation introduce large uncertainties to inventories (Yan et al., 2009; Yan et al., 2003; Zhang et al., 2014). Furthermore, the uncertainty of emission factors related to rice practices is high in China (Peng et al., 2016). In addition, energy activities play an important role in national emissions, with a mean value equal to 24.0 Tg $CH_4$ $yr^{-1}$ and an SD of 2.4 Tg $CH_4$ $yr^{-1}$. Coal mining is the largest emission source, accounting for 77% (18.2 Tg $CH_4$ $yr^{-1}$) of the total energy emissions (Fig. 6a and Table S3). Estimated emissions from coal mining ranged from 16.0 Tg $CH_4$ $yr^{-1}$ in Zhu et al., (2017) to 22.9 Tg $CH_4$ $yr^{-1}$ in NCCC, while estimates from PKU, EDGAR v5.0, and Zhang et al. (2016) showed only a small difference (17.3-19.3 Tg $CH_4$ $yr^{-1}$) (Fig. 6b). EDGAR revised emission factors for coal mining with local data from PKU and weighted the emissions by coal mine activity per province (Janssens-Maenhout et al., 2019). Emissions from waste treatment were 7.4±2.7 Tg $CH_4$ $yr^{-1}$, which contributed a relatively small share of the national total emissions (14%). However, a notable discrepancy exists in emissions from waste treatment, which can be classified into two groups (Fig. 6b). Estimates from PKU, NCCC, GAINS, and Zhang et al. (2016) were 4.3-6.2 Tg $CH_4$ $yr^{-1}$, respectively, while estimates in the others were 8.6-10.4 Tg $CH_4$ $yr^{-1}$ in 2010 (Fig. 6b and Table S3). These differences were mainly induced by the different estimates for wastewater (Table S3). The uncertainty associated with $CH_4$ emissions from wastewater mainly results from the methane correction factor, and the amount of chemical oxygen demand (Peng et al., 2016; Zhao et al., 2019). The high uncertainty in waste emission estimates is generally due to many small point sources and large site-specific variations in emission factors related to different climatic factors and management practices (Höglund-Isaksson, 2012). The detailed regional activity data and localized emission factors used in PKU, NCCC and Zhang et al., (2016) should be taken into account for the variation in local conditions (Table S6-S7).

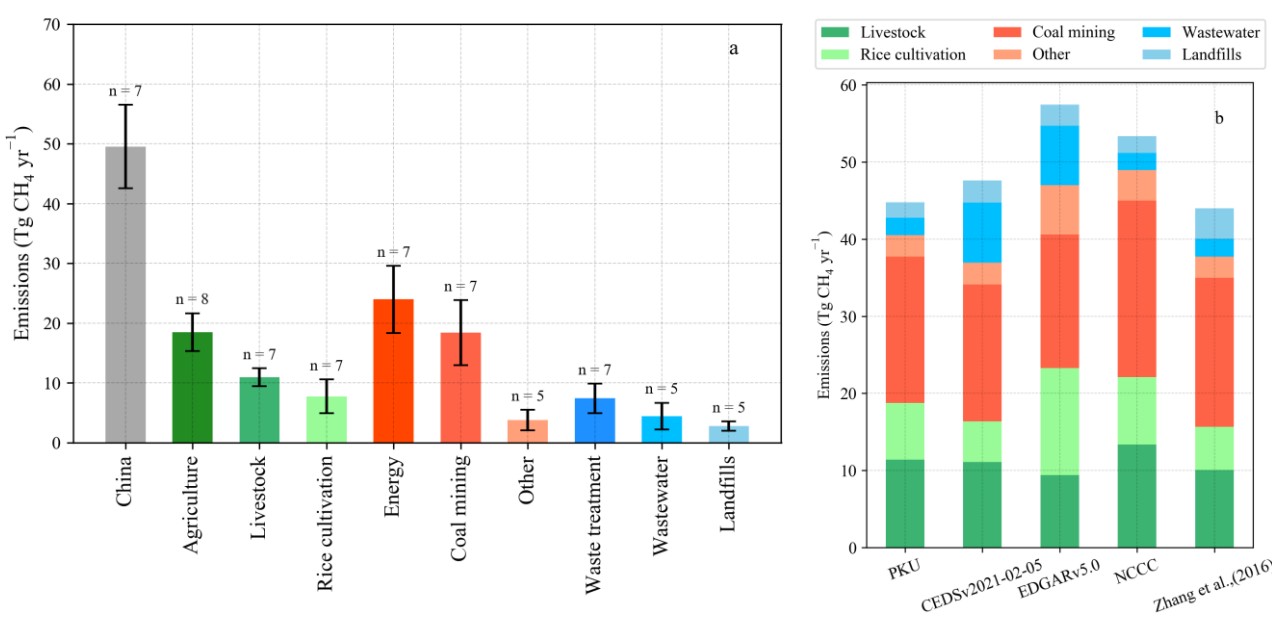

Fig. 6 The mean (bar plot in (a)) and standard deviation (error bar in (a)) of sector and subsector $CH_4$ emissions, and total anthropogenic $CH_4$ emissions by subsector (b) among different inventories in 2010.

## 4 Conclusions

As one of the major rice cultivators and coal producers, China is a large emitter of $CH_4$. Quantifying China's contribution to the global $CH_4$ budget is important and can provide helpful support for policy-making related to mitigating $CH_4$ emissions. We collected and analyzed the currently available datasets to present the amount, uncertainty and spatiotemporal patterns of China's anthropogenic $CH_4$ emissions. Our works shed light on the sources of differences and uncertainties among inventories,Temporally, emissions stabilized in the 1990s but increased significantly thereafter, with AAGRs of 2.6-4.0% during 2000-2010, and slower AAGRs of 0.5-2.2% during 2011-2015. The growth of $CH_4$ emissions is profoundly affected by changes in emissions from the energy sector, with AAGRs of 5.8% - 9.0%. Since 2015, a relatively stable trend was estimated by CEDSv20201-02-05 and our results, with AAGRs of 0.3% and 0.8%, respectively. Spatially, the regional patterns of $CH_4$ emissions were largely associated with economic development and urbanization. Emission hotspots in PKU and EDGAR were mostly located in the North China Plain and south China, which are densely populated areas, energy production regions, and agriculture-dominant regions. Such patterns were not presented in GAINS and REAS, with a lack of emissions hotspots in southern China and biased allocation of the majority of emissions towards Shanxi Province. The incomplete information on emission patterns may mislead or bias mitigation efforts for $CH_4$ emission reductions. During 2000-2010, anthropogenic $CH_4$ emissions from China differed widely among inventories, of which the energy sector contributed the most to the total emissions, followed by agricultural activities, and waste treatment. Large discrepancies are mainly resulted from region-specific activity data and emission factors for coal mining, emission factors for rice cultivation, and emission factors for wastewater. We suggest that data developers should make the detailed activity data for sectors and subsectors publicly available; furthermore, they should use the local optimized emission factors instead of the default emission factors to reduce the level of uncertainty.

Author contributions. XHL and WZ conceived and designed the study. XHL collected and analyzed the data sets. XHL led the paper writing with contributions from all coauthors.

Competing interests. The authors declare that they have no conflicts of interest.

Acknowledgments. This work was supported by the National Key R&D Program of China (No. 2017YFB0504000) and the National Natural Science Foundation of China (No. 41975118). We thank Dr. Steve Smith for kindly provide the latest data of CEDSv2021-02-06 and Ms. Xiaoli Zhou for help in data collection.

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
