# Peer review of "A comparative study of anthropogenic $CH_4$ emissions over China based on the ensembles of bottom-up inventories"

_Earth System Science Data, 2020_

## Referee Comment (RC1) · Anonymous Referee #1 · 30 Sep 2020

Review of "Evaluation of anthropogenic CH4 emissions over China using bottom-up inventories" By Lin X. He et al., submitted to Earth System Science Data

GENERAL COMMENT The authors present a comparison of different bottom-up estimates of methane emissions over China. They discuss the differences in the spatial distributions and temporal changes in methane emissions for total anthropogenic emissions and some sectoral emissions. They use both global and regional data set to feed the discussion. They conclude on methane emission changes in China over the past decade and on the uncertainties on estimating methane emissions.

The manuscript is readable and intelligible. However English polishing would be welcome (use of past tense during the discussion of the results seems odd). The figures are hardly legible (small font, pale colours) and need to be revised. Also, the study is

lacking more detailed information on the data set used, and on the differences between inventories explaining discrepancies in methane amount or changes. Some information is provided in the supplementary but never mentioned or used in the main text. Some data sets are not introduced properly (sources of data/methods). The title could suggest that the BU estimates are evaluated, which is not the case. They are only (poorly) compared to each other. The reader would expect to gain insights on which data set provide the best estimates (depending on the sector maybe). Are some data sets out-dated for some reason (REAS?)? Why? What kind of activity data / emissions factors are the best to be used to properly represent Chinese emissions? At the end of the manuscript, the reader has learned a bit on Chinese emissions (sector contribution, trends, some uncertainties) – assuming he did not know anything, but he doesn't know what to do with these different data sets, nor what is the best suited for any specific sector. However, such inventories comparison could be useful to the community, especially to highlight existing regional/sector specific data sets, under the conditions that thorough presentation, analysis and evaluation of the inventories are performed. Lastly, evaluation of inventories could be done using regional modelling of methane concentrations over China. For the above reason, I would recommend publication in ESSD but only after major revisions addressing the general and specific comments highlighted in this review. Below are some comments to help improving the manuscript and study, but more comments could have been risen.

SPECIFIC COMMENTS

Section 1 – Introduction

Line 36. Might be worse specifying that the value is a global mean average over remote marine stations (I guess. . .)

Line 48-49. The reasons are not well explained here. Please detail. There are numerous estimates of methane emissions (as used in this study), the (small?) number of estimates is not the reason of the challenge of methane compared to CO2. The

challenge comes from the processes leading to methane emissions (leaks, biogenic emissions, ...) that are far much uncertain and difficult to estimate than emissions from combustion of fossil fuels – LULUCF standing aside.

Line 53. This sentence is omitting regional inversions at higher resolution (for example, Thompson et al., 2015, https://doi.org/10.1002/2014JD022394). If inversions are discussed here broader and appropriate literature needs to be added.

Line 58. "the quality" of activity data. The sentence seems to say that the activity data are of good quality – positive; while they are known to be quite uncertain.; the whole sentence is too long and lacks of specific criticism against activity data and or emissions factors. Are they both uncertain in space and time? Split the sentence and rephrase to explain better why "the use of BU is challenging " (I guess "challenging in modelling studies"... not specified).

Line 62. I do not agree. Many studies have discussed Chinese methane emissions. Many of them are cited in this ms, and probably many other exist. Most of the global methane studies do mention Chinese emissions, their amount, spatial distribution, changes over time, and even uncertainties on emissions factor, specifically related to coal. This sentence is not above China, but more about the attention given to methane compared to carbon dioxide. Please rephrase or delete or move up.

Line 66. Sort sectors by order of importance: energy is first with 45%!

Line 67-68. "large part of the variability". Variability of what?

Line 72. What could be the reasons explaining these differences? Paddy areas? Parametrization? Cultivation practices hypothesis?

Line 74. Meaning that there is none on solid waste?

Line 78. What is meant by "systematic"? is it regularly? Or with common procedure across sectors?

[Figure]

Line 79 "Further mitigate climate warming": not clear, please rephrase

Lines 80-81: The author state that they collect most of the existing data sets. This means that some are missing? And could have been included as well? It is stated here that 12 global and regional inventories have been gathered? Do they all covered all sectors? Table 1 lists only 4 global inventories. . .

Section 2 – Data and Methods

Lines 89-100. Following the previous comment. Table 1 lists only the 4 gridded data set. No detail is given regarding the 8 so-called "statistical data sets". What is "statistical datasets"? After stating the number of inventories, sectors are introduced, then we go back to the inventories. Please reorganize the section. The PKU inventory includes all anthropogenic sectors apparently and is not just a " fuel combustion inventory" The Global Methane Budget is based on EDGARv432- EPA- ECLIPSE v6 – FAO (agriculture) – CEDS - / so there is some overlap with the other datasets used here. The reader has no idea about the "published literature Yue, Huang, Zhang and Chen and Zhang2016 Zhang2018. A table and/or text are needed to explain these data sets: sectors/methods/ specificity. Why are this specific dataset are included? What is their added-value? How do the reader access to them? (it is mandatory to have this information)

Line 102. Why 3 and not gridded data sets here? this sentence is true for any inventory, gridded or country based.

Line 104. What is the point stating that some inventories are used in inversion? It's really depending on the group doing the inversion (global..?).

Section 3 Results and discussions.

Line 111-112. This is not really surprising as the Global Methane Budget is based -approximately on the same data set.

Section 3.1 is entitled "temporal variations of anthropogenic emission" but half of it

compares the magnitude of the inventories emissions or the relative contribution of the sectors. Please organize the discussion.

Line 115 – Energy sectors dominates (27-60%) but then agriculture activities contribute to 27-50%... is the relative contribution varying across inventories? Over time? does it means that the splitting is much different from one inventory to another? a stacked bar plot could be more explicit. Fig 1 – revised the colors and font size of the fig. FAO data are shown for all sectors. The authors have to acknowledge the source of FAO data for other sectors than agriculture. FAO produces only agriculture emissions; other sectors are from third party.... After digging a bit, it comes from PRIMAP hist. dataset v2.1 (Gutshow 2016)... using EDGARv42 for the recent period. No information is given regarding this dataset.

Line 119-121: what is the link between REAS and GOME data? Here again lack of description of the data set prevent the reader to understand the suggested conclusion.

Line 123. "these results may be due .." such a study need to be more persuasive and provide better and thorough explanations on the differences or discrepancies between inventory?

Line 124/125. What is NCCC? Is there any ref to cite "proving" what is stated about lower emissions factors?

Line 128. Here and elsewhere, when discussing EDGAR please specify the version, the trend will differ from one version to another. FAO other sector being based apparently on EDGARv42... EPA is EPA2012 (projection form 2005?) this need to be specified somewhere

Line 131: "may be caused by". See previous comment, the readers need to be more confident in the results. Data/publication that could strengthen this suggestion/conclusion? Line 132 : subsectors are discussed but no plot are shown in the main text nor in the supplementary, how the reader can check the findings?

Line 147: The authors states that the gridded emissions are limited. But some are missing here. CEDS (based on EDGARv42) is missing. There are other version of EDGAR that could have been integrated (V432) for comparison.

Line 148-150. Again relative contribution of sector at national scale is discussed here... reorganize please. (Are the number in agreement with the previous ones?)

Line 150 and following. Discussing province differences could be difficult to follow for non-specialist of Chinese province. Please add a map (in the supplementary) of the provinces.

Line 164. Emissions factors control more the magnitude than the spatial distribution, which is more related to activity data used to spatialized the emissions.

Line 169 : What is NBS?

Line 170 and below: Here the reader is waiting for information on the sources of emission factors, activity data and specialization hypotheses that may differ between inventories explain the discrepancies in different sectors. Could the different resolution of the inventory different induce visual spatial differences?

Fig2 and 3. The figures are hardly ligible. Color scale/ names of the inventories too small. Fig 2 right panels: the reader cannot read this. When is this useful? What is the grid resolution of these plots? EDGAR spatial distribution (and trend in Fig3) for energy is really different from the other inventories. Is such a distribution realistic?

Line 190-195. Trend in agriculture emissions. Why EDGAR is so different from the other? Are this conclusions in agreement with FAO ? with more specific agricultural?

Line 225-230. Here the authors state that rice cultivation areas (emissions have increased) ; while previously (line ca 190), a decrease in emission from rice cultivation was presented. How is that consistent?

Line 230. Livestock includes enteric fermentation+ manure management? There

is a recent paper on enteric fermentation that could be useful (Chang et al., 2019, https://www.nature.com/articles/s41467-019-11066-3)

Line 232; "REAS underestimates emissions" compared to what? Is there a reference/preferred inventory taken as reference?

Line 235(around). Here a discussion is needed on the sources of activity data used in the different inventories. Most of them use FAO, aren't they? So why such differences?

Lien 247. This could be easily avoided by plotting the maps on the same resolution. See also a previous comment.

Fig 4. Same comments as for other figures. What are the horizontal resolutions of the maps? Do they differ from one inventory to another?

Line 253 and following. "considering the comparability of different inventory"; This is not clear. REAS seems to have been excluded from the mean calculation. Is there a reason? EPA is included while this seems to be EPA2012 data set, which is a projection from 2005 onward. How such a data set is valid? These mean/SD calculations seem to be the concluding numbers of this study, though they use a subset of the dataset discussed. What are the selection criteria? Is that the "best guess for Chinese methane emissions"? are the considered inventories the best ones for all categories?

Line 268-272. Two groups of inventories are formed based on waste emissions. Which group is the most realistic? Can we conclude on this?

Section 4 Conclusions. The end of the conclusion mentions the used of default emissions factor instead of province specific emissions factors. This is the first time this point is mentioned and more discussion is needed on that throughout the paper. Are inventories using default emissions factor as reliable as others? More constructive criticism of the methodology used in the different inventories is needed to assess which inventory fits the best the reality.

TECHNICAL COMMENTS

Line 37. Remove "However" Line 48 Change to "contribute the most to global . . .." Line 61 Change "area" to "country" Line 135 : substantial variability. Do you mean increase?

---

## Referee Comment (RC2) · Anonymous Referee #2 · 27 Oct 2020

This manuscript compared several bottom-up CH4 emission inventories, investigated the spatial-temporal patterns of CH4 emissions in China, and tried to explain the discrepancies between different data products to evaluate those emission inventories. However, I think the authors did not successfully achieve their research aim.

First, the comparison between different inventories did not provide any more information on the CH4 emission characteristics in China. The previous bottom-up emission inventories have already presented what this study has shown here. This research has produced very few new findings.

Second, the authors just compared the emission values between different inventories, however, such a simple analysis and comparison cannot provide us an evaluation of anthropogenic CH4 emissions inventories (mentioned by the paper title). Many explanations of the discrepancies between different inventories did not provide any evidence and cannot fully convince me.

Third, the analysis of emission spatial distributions did not make sense because the global and regional inventories listed in Table 1 typically allocated the country- and province-level emission estimates to grid cells to create emission maps. Some of the selected spatial allocation proxies are rather arbitrary in my opinion, which cannot provide us accurate emission mapping results. Therefore, the comparisons shown in Figures 2 to 4 cannot give us any useful information.

Finally, I do not agree that the authors said "This study, to the best of our knowledge, provides the first quantitative analysis of the amount and spatiotemporal patterns of CH4 emissions in China" in the conclusion section because of my comments above.

Overall, I don't see much scientific significance in this paper though it summarized plenty of data and did some analysis. The paper is not well written and needs lots of editing. I fear I cannot recommend this paper for publication in its current form.
* * *

---

## Author Comment (AC1) · 16 Dec 2020

Reviewer 1#

GENERAL COMMENT The authors present a comparison of different bottom-up estimates of methane emissions over China. They discuss the differences in the spatial distributions and temporal changes in methane emissions for total anthropogenic emissions and some sectoral emissions. They use both global and regional data set to feed the discussion. They conclude on methane emission changes in China over the past decade and on the uncertainties on estimating methane emissions. The manuscript is readable and intelligible. However English polishing would be welcome (use of past tense during the discussion of the results seems odd). The figures are hardly legible

(small font, pale colours) and need to be revised. Also, the study is lacking more detailed information on the data set used, and on the differences between inventories explaining discrepancies in methane amount or changes. Some information is provided in the supplementary but never mentioned or used in the main text. Response: We thank the reviewer for understanding the merits of our work. We also appreciate the reviewer for her/his constructive comments, which has significantly improved the quality of the paper. We addressed these comments point-by-point as follows. According to the reviewer's comments, in the revised paper, the language and editting has been polished by a native editor, and we have changed the past tense to present tense for the discussion part (e.g. in Lines 212, 213, 298, 306, 307, 331). Larger fonts and brighter colors have been adopted in producing the figures, which makes these figures more readable. Moreover, we added a detailed Table S1 summarizing the data sets used in this study, and we also included the changes of sectoral emissions by introducing Figures S2 and S3 to explain source of difference. More descriptions on explaining discrepancies in methane amount and changes among inventories (Lines 157-162) have also been included in the revision. In this revision, all the supplementary files are correctly referred in the main text. Some data sets are not introduced properly (sources of data/methods). The title could suggest that the BU estimates are evaluated, which is not the case. They are only (poorly) compared to each other. The reader would expect to gain insights on which data set provide the best estimates (depending on the sector maybe). Are some data sets out-dated for some reason (REAS?)? Why? Response: Good point! In the revision, we have included more descriptions/details on these inventories in the Methods section (Lines 110-131). Firstly, we changed "Evaluation" to "Comparison" in the title. Besides, we have added a detailed Table S1 to summarize all the data sets used in our study. Moreover, we have added more evaluating descriptions, e.g. in Lines 203-206 and Lines 216-226. For magnitude estimates, the NCCC could be considered as a reference, those have lager differences may need further check and should be cautiously used. We have clarified this in the revised manuscript (Lines 310-313). Moreover, we have added the kappa statistics in

sect 3.2 to quantatively identify the difference among inventories. Among the datasets used in this study, REAS2.1 is an old version since the newest version (REAS3.2) stopped CH4 update, and the reasons may be this inventory concentrated more on polluted gases (http://www.nies.go.jp/REAS/index.html#data%20sets) (Kurokawa and Ohara, 2020). Other data sets such as Yue et al., (2012) did not update too, which may be due to lack of human or funding resources. What kind of activity data / emissions factors are the best to be used to properly represent Chinese emissions? Response: Since CH4 emissions has large spatial-temporal variations and also the complexity of the processes included. To reduce uncertainties in emission estimates, detailed local to regional specific emission factors and temporal dynamic activity data are properly to produce more accurate inventory. We added discussions on this topic in lines 177 for coal emissions, 310-313 for rice emissions, 364-365 for localized emission factors (EF) (Table S6-S7). Using coal as an example, we recommended provincial EF and activity data from Sheng et al (2019). At the end of the manuscript, the reader has learned a bit on Chinese emissions (sector contribution, trends, some uncertainties) – assuming he did not know anything, but he doesn't know what to do with these different data sets, nor what is the best suited for any specific sector. Response: Thank the reviewer for pointing out this. As descried for the above question, we added recommendations for the use of these inventories. We added more suggestions on the use of these inventories, e.g. in Lines around 310-313 for rice cultivation emissions, and 359-365 for waste emissions. In total amount estimates, EDGAR and FAO are close to NCCC, but FAO's close in total is resulted from higher estimates in energy and lower estimate in agriculture. Overall, we suggest the use of sector specific inventories with provincial activity data and EFs for improvements of national-data-based inventories. Specifically, for rice cultivation, we recommend Zhang et al., (2017) for the detailed regional water management data and provincial organic matter application rates, which is a Tier 3 method used in NCCC as part of national inventory reported to UNFCCC (Lines 310-313). For livestock fermentation, the regional temperature-dependent and species specific EFs from PKU are better than the default ones. For coal mines emissions, we

suggest that estimates from Sheng et al., (2019) have more reliable emission patterns by using more than 10000 coal mines and the provincial activity data in coal productions. For waste treatment, emissions estimate from PKU which are based on the provincial statistics are more reliable, and show a minor difference with the reference values from NCCC. And if for atmospheric transport or inversion studies, total emissions covering full time-series is more important and PKU is recommended. We have thoroughly addressed this issue in the revised MS (Lines 310-313, 364-365), we hope this can satisfy the reviewer's concerns. However, such inventories comparison could be useful to the community, especially to highlight existing regional/sector specific data sets, under the conditions that thorough presentation, analysis and evaluation of the inventories are performed. Lastly, evaluation of inventories could be done using regional modelling of methane concentrations over China. For the above reason, I would recommend publication in ESSD but only after major revisions addressing the general and specific comments highlighted in this review. Response: We thank reviewer for her/his positive comments of our study. Following her/his suggestion, we have added more discussions on the evaluation of inventories using modelling methods.

Below are some comments to help improving the manuscript and study, but more comments could have been risen. Response: Thank you for your careful review and editing, we have addressed all your concerns and details are shown below.

SPECIFIC COMMENTS Section 1 – Introduction Line 36. Might be worse specifying that the value is a global mean average over remote marine stations (I guess...) Response: Revised accordingly in Lines 39-40. Line 48-49. The reasons are not well explained here. Please detail. There are numerous estimates of methane emissions (as used in this study), the (small?) number of estimates is not the reason of the challenge of methane compared to CO2. The challenge comes from the processes leading to methane emissions (leaks, biogenic emissions, ...) that are far much uncertain and difficult to estimate than emissions from combustion of fossil fuels – LULUCF standing aside. Response: Thank you and we stand with you on the

comment that the large uncertainty in CH4 comes more from the processes. We have revised accordingly in the revision (Lines 52-56). Line 53. This sentence is omitting regional inversions at higher resolution (for example, Thompson et al., 2015, https://doi.org/10.1002/2014JD022394). If inversions are discussed here broader and appropriate literature needs to be added. Response: Modified accordingly (Lines 61-63).

Line 58. "the quality" of activity data. The sentence seems to say that the activity data are of good quality – positive; while they are known to be quite uncertain; the whole sentence is too long and lacks of specific criticism against activity data and or emissions factors. Are they both uncertain in space and time? Split the sentence and rephrase to explain better why "the use of BU is challenging " (I guess "challenging in modelling studies": : : not specified). Response: Modified accordingly (Lines 70-73).

Line 62. I do not agree. Many studies have discussed Chinese methane emissions. Many of them are cited in this ms, and probably many other exist. Most of the global methane studies do mention Chinese emissions, their amount, spatial distribution, changes over time, and even uncertainties on emissions factor, specifically related to coal. This sentence is not above China, but more about the attention given to methane compared to carbon dioxide. Please rephrase or delete or move up. Response: Deleted accordingly (Line 75-77).

Line 66. Sort sectors by order of importance: energy is first with 45%! Response: Revised accordingly (Line 79).

Line 67-68. "large part of the variability". Variability of what? Response: It means the difference among inventories. We have revised (Line 81).

Line 72. What could be the reasons explaining these differences? Paddy areas? Parametrization? Cultivation practices hypothesis? Response: Thank you for asking. On average, parametrization and model imperfection contributed 56.6% of the uncertainty, while errors and the scarcity of input data (irrigation and organic matter

input amount) contributed the rest. We have clarified this in the revision (Lines 85-87).

Line 74. Meaning that there is none on solid waste? Response: Thank you for pointing out this. We have included those on solid waste in the revision (Lines 90-91).

Line 78. What is meant by "systematic"? is it regularly? Or with common procedure across sectors? Response: Here we mean comprehensive, not regularly, but in a systematic way (common data process procedures) for all inventories and sectors. We have clarified these accordingly in Lines 94-95.

Line 79 "Further mitigate climate warming": not clear, please rephrase Response: Modified accordingly in Line 96.

Lines 80-81: The author state that they collect most of the existing data sets. This means that some are missing? And could have been included as well? It is stated here that 12 global and regional inventories have been gathered? Do they all covered all sectors? Table 1 lists only 4 global inventories. Response: Thank you for your helpful suggestion; following which, CEDS has been introduced in the revision. We have included all the datasets available to our best knowledge and ability in this study (Lines 97-98). We have added the detail information of inventories (8 tabular data) in Table S1. Table 1 listed the gridded data sets used in MS. All the 13 inventories covered all sectors. The sector information is listed in Table S2.

Section 2 – Data and Methods Lines 89-100. Following the previous comment. Table 1 lists only the 4 gridded data sets. No detail is given regarding the 8 so-called "statistical data sets". What is "statistical datasets"? After stating the number of inventories, sectors are introduced, then we go back to the inventories. Please reorganize the section. The PKU inventory includes all anthropogenic sectors apparently and is not just a " fuel combustion inventory" The Global Methane Budget is based on EDGARv432- EPA- ECLIPSE v6 –FAO (agriculture) – CEDS - / so there is some overlap with the other datasets used here. The reader has no idea about the "published literature Yue, Huang, Zhang and Chen and Zhang2016 Zhang2018. A table and/or text are needed

to explain these data sets: sectors/methods/ specificity. Why are this specific dataset included? What is their added-value? How do the reader access to them? (it is mandatory to have this information) Response: We added detailed information of inventories (8 tabular data) in Table S1 and Table S2 including activity data, sector and sources. Thank you for the reorganization suggestion. We moved the sectors introduction after the 13 inventories descriptions. We removed the "fuel combustion inventory" for PKU, and indicated that GMB included some data used in this study. We introduced other published literatures in Table S1 and acknowledged the data availability in the main text (in Data and Methods part). The added value of published literatures is cross-check of inventories using independent studies to include potential uncertainties and enhance the confidence of estimates. Some of the 8 tabular published literatures used provincial activity data and published in high-impact journals (e.g. Zhang and Chen, 2014).

Line 102. Why 3 and not gridded data sets here? this sentence is true for any inventory, gridded or country based. Response: We add the general information for explaining the classification of two groups in Lines around 135.

Line 104. What is the point stating that some inventories are used in inversion? It's really depending on the group doing the inversion (global..?). Response: Here, we meant to mention that one of the functions of inventory is used as prior for atmospheric inversion. And which inventory is used in the inversion model depend on the international fame, suitability (spatial-temporal coverage) of inventories for individual studies. CEDS, EDGAR and GAINS are more well-known data, and used more widely in atmosphere inversion.

Section 3 Results and discussions. Line 111-112. This is not really surprising as the Global Methane Budget is based-approximately on the same data set. Response: We agree with the reviewer and added more descriptions on annual changes of used datasets and also differences with GMB accordingly in Lines 155-156. Section 3.1 is entitled "temporal variations of anthropogenic emission" but half of it compares the magnitude of the inventories emissions or the relative contribution of the sectors.

[Figure]

Please organize the discussion. Response: Thank you for these comments, and we added more descriptions on temporal variations and deleted sectoral descriptions in Lines 157-163. Line 115 – Energy sectors dominates (27-60%) but then agriculture activities contribute to 27-50%... is the relative contribution varying across inventories? Over time? Does it mean that the splitting is much different from one inventory to another? a stacked bar plot could be more explicit. Fig 1 – revised the colors and font size of the fig. FAO data are shown for all sectors. The authors have to acknowledge the source of FAO data for other sectors than agriculture. FAO produces only agriculture emissions; other sectors are from third party? After digging a bit, it comes from PRIMAP hist. dataset v2.1 (Gutshow 2016) using EDGARv42 for the recent period. No information is given regarding this dataset. Response: Yes, as the reviewer suggested: sectoral relative contribution varies across inventories and over time. We revised the sector contribution to temporal variation of methane emissions in Line 157. Figure S2 added for staked bar plot of sectoral contributions over time. We revised the colors and font size accordingly in Figure 1. Thank you. We added FAO data description in Table S1. Line 119-121: what is the link between REAS and GOME data? Here again lack of description of the data set prevent the reader to understand the suggested conclusion. Response: We revised in Lines 169-170. The link between REAS and GOME data is that GOME trend served as an independent verification for activity data. After verification with higher GOME trend (increased 50% from 1996-2002) than provincial statistical trend (25%) and IEA trend (15%), statistical data on coal consumption were modified to higher values in the China Statistical Yearbook.

Line 123. "these results may be due .." such a study need to be more persuasive and provide better and thorough explanations on the differences or discrepancies between inventory? Response: Indeed, this was mainly caused by higher EF for EDGAR than NCCC. We revised in Lines . For coal mining, EDGAR EF is 10.0 m3/t, while NCCC is 8.89 m3/t (Table S4); For rice cultivation, EDGAR EF is 0.1-1.4 g/m2/d, while NCCC is 0.005-0.21 g/m2/d (Table S4).

Line 124/125. What is NCCC? Is there any ref to cite "proving" what is stated about lower emissions factors? Response: NCCC is the National Communication on Climate Change (NCCC) of the People's Republic of China, which is considered as official data for China's GHG emissions. We added the EF of NCCC in Table S4, and added the reference. Line 128. Here and elsewhere, when discussing EDGAR please specify the version, the trend will differ from one version to another. FAO other sector being based apparently on EDGARv42? EPA is EPA2012 (projection form 2005?) this need to be specified somewhere Response: Thank you for this good suggestion. We specified version information for EDGAR. and FAO other sectors are taken from the third-party PRIMAP-hist dataset v2.1 (Gütschow et al., 2016; Gütschow et al., 2019), and we listed them in Table S1. EPA 2012 is removed due to the reasons of projection from 2005, and detailed information provided in Table S1. Line 131: "may be caused by". See previous comment, the readers need to be more confident in the results. Data/publication that could strengthen this suggestion/conclusion? Response: Indeed, we checked the publication and data, and drew conclusions with more confidence. Line 132 : subsectors are discussed but no plot are shown in the main text nor in the supplementary, how the reader can check the findings? Response: Thank you, we added subsector plot in Fig. S3. Line 147: The authors state that the gridded emissions are limited. But some are missing here. CEDS (based on EDGARv42) is missing. There are other version of EDGAR that could have been integrated (V432) for comparison. Response: Thank you for this suggestion. We added CEDS dataset in Figure and more discussions on EDGAR older versions, but to keep simplicity, we used the newest EDGAR version in the main text. Line 148-150. Again relative contribution of sector at national scale is discussed here... reorganize please. (Are the number in agreement with the previous ones?) Response: Thank you. We kept the relative contribution of sector at national scale here and deleted the previous ones. Yes, this is consistent with previous ones, this is one year (2010) and falls into the previous larger ranges. Line 150 and following. Discussing province differences could be difficult to follow for non-specialist of Chinese province. Please add a map (in the supplementary) of the provinces. Response: Thank

you, and we added the province map of Fig. S1 in the supplementary.

Line 164. Emissions factors control more the magnitude than the spatial distribution, which is more related to activity data used to spatialized the emissions. Response: Thank you for pointing out this, we revised in Line 230-231.

Line 169 : What is NBS? Response: NBS is National Bureau of Statistics of China, and we added full name for abbreviation in Table 1 and Data and Methods in Line 140. Line 170 and below: Here the reader is waiting for information on the sources of emission factors, activity data and specialization hypotheses that may differ between inventories explain the discrepancies in different sectors. Could the different resolution of the inventory different induce visual spatial differences? Response: Thank you, we revised in Line 240-244. We changed the total emissions per grid cell to emission intensity (g CH4 km-2) and thus the different resolutions of the inventories would not induce visual spatial differences.

Fig2 and 3. The figures are hardly legible. Color scale/ names of the inventories too small. Fig 2 right panels: the reader cannot read this. When is this useful? What is the grid resolution of these plots? EDGAR spatial distribution (and trend in Fig3) for energy is really different from the other inventories. Is such a distribution realistic? Response: Thank you, we revised Fig. 2 and 4, enlarged the color scale and names. Font size in Fig 2 right panels were enlarged and used in Lines 236-239 and 244 (Fig. 2v) to discuss the differences of spatial distribution (e.g. EDGAR distribution features). The grid resolution of these plots was original datasets resolutions in 0.1°or 0.25°(Added in figure caption). EDGAR originally uses 328 coal mines with locations for China from world coal association (https://eerscmap.usgs.gov/wocqi) as point emissions to disaggregate the amount of national emissions (Greet et al., 2019), and then update by Liu et al. (2015). However, emissions from coal mining estimated by EDGAR still have notable bias toward Shanxi province (Fig. 5f), while PKU and Sheng et al., 2019 used independent sources of 4264 and over 10,000 coal mines. As a result, EDGAR put small coal mines emissions to large coal mines.

Fig. 1 The spatial location of coal mines in China from world coal association (a), spatial distribution of methane emissions from coal mines in EDGAR (b).

Line 190-195. Trend in agriculture emissions. Why EDGAR is so different from the other? Are these conclusions in agreement with FAO? with more specific agricultural? Response: We added the temporal variation of emissions from rice production and livestock (Fig. S3). The decreasing trend in southern China for EDGAR is consistent with GAINS and some provinces in PKU (Fig. 4 in the main text), while FAO only provides tabular data and no gridded maps, the trend comparison with FAO is not possible. For livestock, EDGAR and FAO show a similar trend (Fig. S3). For rice production, EDGAR is consistent with FAO (and other 4 datasets) with two dips in 2003 and 2007 (Fig. 2).

Fig. 2 The temporal variation of emissions from rice cultivation

Line 225-230. Here the authors state that rice cultivation areas (emissions have increased); while previously (line ca 190), a decrease in emission from rice cultivation was presented. How is that consistent? Response: The previously statement of decreasing emissions are mainly due to adopt the practice of draining paddy fields in the middle of the rice-growing season (Qiu 2009). The combined effects of the decrease in irrigation management and area expansion are decreasing emissions.

Line 230. Livestock includes enteric fermentation+ manure management? There is a recent paper on enteric fermentation that could be useful (Chang et al., 2019, https://www.nature.com/articles/s41467-019-11066-3) Response: Yes. We added this reference in the discussion (Lines 314). Livestock emissions include enteric fermentation and manure management. According to Chang et al., (2019), enteric fermentation from ruminants dominates the livestock emissions, and manure management has a smaller contribution.

Line 232; "REAS underestimates emissions" compared to what? Is there a reference/preferred inventory taken as reference? Response: Here, it means lower estimate than all the other inventories. Revised in Line 317. NCCC is generally considered as a reference and Lin et al., (2011) is the closest one.

Line 235(around). Here a discussion is needed on the sources of activity data used in the different inventories. Most of them use FAO, aren't they? So why such differences? Response: Thank you, we added it in the discussion for rice production in Lines 310-313. The spatial distribution is strongly correlated with the proxy data. National and sub-national FAO livestock statistical data are made into gridded maps using machine learning method (Gilbert et al., 2018), and PKU used province-level annual census data from agriculture statistics yearbooks (China Agricultural Statistical Yearbook, 1980–2010). We communicated with Gilbert and point out this large difference in Beijing, he explained that this could be resulted from the proxy method. Add the information in Line 321. Lien 247. This could be easily avoided by plotting the maps on the same resolution. See also a previous comment. Response: The gridded maps of REAS and Sheng et al., (2019) are based on a spatial resolution of 0.25 degree, while others are in 0.1 degree, to keep the higher resolution and make them comparable, emission intensity are calculated and showed. Emissions from coal mining in EDGAR v5.0 are not clear in 0.1 degree, but the bar chart is calculated using its original spatial resolution to obtain the original value, and thus would not change the result.

Fig 4. Same comments as for other figures. What are the horizontal resolutions of the maps? Do they differ from one inventory to another? Response: Revised accordingly. The horizontal resolutions of the maps are shown within Fig.2 (under inventory name). The emissions intensity has considered the area of grid cells and thus is comparable. Line 253 and following. "considering the comparability of different inventory"; This is not clear. REAS seems to have been excluded from the mean calculation. Is there a reason? EPA is included while this seems to be EPA2012 data set, which is a projection from 2005 onward. How such a data set is valid? These mean/SD calculations seem to be the concluding numbers of this study, though they use a subset of the dataset discussed. What are the selection criteria? Is that the "best guess for Chinese

methane emissions"? are the considered inventories the best ones for all categories? Response: Thank you, here "comparability" means the same year (2010), completeness of all same subsectors (added in Lines 340), REAS was excluded due to lack of 2010 data. We excluded the EPA2012 data. The selection criteria for total emissions is the same year (2010), and completeness of all same subsectors. While for subsectors we included studies focused on only subsectors. Since we included estimates as much as possible and are likely to eliminate systematical errors, which make it a potential "best guess". The considered inventories are well known ones representing the state of the art estimates. Line 268-272. Two groups of inventories are formed based on waste emissions. Which group is the most realistic? Can we conclude on this? Response: Thank you for this question and we are tended to think group 1 is more realistic due to the detailed provincial activity data and localized EF they used while group 2 used national activity data and IPCC default EF. Section 4 Conclusions. The end of the conclusion mentions the used of default emissions factor instead of province specific emissions factors. This is the first time this point is mentioned and more discussion is needed on that throughout the paper. Are inventories using default emissions factor as reliable as others? More constructive criticism of the methodology used in the different inventories is needed to assess which inventory fits the best the reality. Response: Thank you, we added in Lines around 177-178, 310-313, 363-365. Peng et al., (2016) had compared their results (PKU) with the estimates using same method but IPCC default emission factors (IPCC-EF). Estimates from PKU are consistent with IPCC-EF, but ∼30% lower after 2000. We suggest the use of the best available information (e.g. regional-specific activity data and emission factors for each source sector, which is an important way forward in improving accuarcy of CH4 emission inventories.

TECHNICAL COMMENTS Line 37. Remove "However" Response: Revised accordingly.

Line 48 Change to "contribute the most to global. . .." Response: Thank you. Revised accordingly.

Line61 Change "area" to "country" Response: Revised accordingly in Line 74.

Line 135: substantial variability. Do you mean increase? Response: Revised accordingly in Line 186.

Reference Andrew, R. M.: A comparison of estimates of global carbon dioxide emissions from fossil carbon sources, Earth System Science Data, 12, 1437-1465, 2020. Höglund-Isaksson, L.: Global anthropogenic methane emissions 2005-2030: technical mitigation potentials and costs, Atmospheric Chemistry and Physics, 12, 9079-9096, 2012. Han, P., Zeng, N., Oda, T., Lin, X., Crippa, M., Guan, D., Janssens-Maenhout, G., Ma, X., Liu, Z., and Shan, Y.: Evaluating China's fossil-fuel CO 2 emissions from a comprehensive dataset of nine inventories, Atmospheric Chemistry and Physics, 20, 11371-11385, 2020. Kurokawa, J. and Ohara, T.: Long-term historical trends in air pollutant emissions in Asia: Regional Emission inventory in ASia (REAS) version 3, Atmospheric Chemistry and Physics, 20, 12761-12793, 2020. Lin, Y., Zhang, W., and Huang, Y.: Estimating spatiotemporal dynamics of methane emissions from livestock in China, Environmental Science (in Chinese), 32, 2212-2220, 2011. Liu, Z., Guan, D., Wei, W., Davis, S. J., Ciais, P., Bai, J., Peng, S., Zhang, Q., Hubacek, K., and Marland, G.: Reduced carbon emission estimates from fossil fuel combustion and cement production in China, Nature, 524, 335-338, 2015. Saunois, M., Stavert, A. R., Poulter, B., Bousquet, P., Canadell, J. G., Jackson, R. B., Raymond, P. A., Dlugokencky, E. J., Houweling, S., and Patra, P. K.: The global methane budget 2000-2017, Earth System Science Data, 2020. doi: 10.5194/essd-5112-1561-2020, 2020. Sheng, J., Song, S., Zhang, Y., Prinn, R. G., and Janssens-Maenhout, G.: Bottom-up estimates of coal mine methane emissions in China: a gridded inventory, emission factors, and trends, Environmental Science & Technology Letters, 6, 473-478, 2019. Zhang, W., Sun, W., and Li, T.: Uncertainties in the national inventory of methane emissions from rice cultivation: field measurements and modeling approaches, Biogeosciences, 14, 163-176, 2017.

Please also note the supplement to this comment:
https://essd.copernicus.org/preprints/essd-2020-210/essd-2020-210-AC1-
supplement.pdf

---

## Author Comment (AC2) · 16 Dec 2020

Reviewer 2# This manuscript compared several bottom-up CH4 emission inventories, investigated the spatial-temporal patterns of CH4 emissions in China, and tried to explain the discrepancies between different data products to evaluate those emission inventories. However, I think the authors did not successfully achieve their research aim. First, the comparison between different inventories did not provide any more information on the CH4 emission characteristics in China. The previous bottom-up emission inventories have already presented what this study has shown here. This research has produced very few new findings. Response: Thank you for your comments. Indeed, we stand with you that several individual studies have been done in emission estimates for global and reginal studies (e.g. EDGAR, REAS, CEDS, and PKU, etc.) and here we

are not to provide a new dataset in this study. However, there remain significant merits in our study, which will mainly benefit the research community. First, existing individual estimates (e.g. bottom-up approach) exhibited wide ranges due to the complex emitting processes, large amount of activity data, and various site-specific emission factors (Höglund-Isaksson, 2012; Saunois et al., 2020). While "true" emissions cannot be known, by comparing different datasets can enable identification of the reasons for those disparities and sources of uncertainties (Andrew, 2020). This is an important way forward in improving accuracy of CH4 emission inventories. To the best of our knowledge, such comparisons have seldom been conducted, particularly for CH4 emissions across different sectors at the national scale of China. Second, we have actually presented the new findings by gathering all the publicly available emission dataset. For example, we quantitatively compared all the collected estimates and explored the reasons for differences both in amount and spatial-temporal pattern. We provided a state-of-the-art mean and uncertainty estimates of national total and sectoral CH4 emissions (Fig. 5 and Table S3). Our results reveal that REAS is a potential outlier, which presents an abnormal increasing trend of China's CH4 emissions (Fig. 1). Further, the global inventories like CEDS and EDGAR are widely used as priori emissions for atmospheric transport or assimilation researches. However, emissions from coal mining estimated by EDGAR show notable bias toward Shanxi province (Fig. 4f). Emissions from energy sector in CEDS show an extremely higher estimates of 40 Tg CH4 in 2010, while other inventories are within ranges of 22-27 Tg CH4 (Table S3). The uncertainty in priori information will bias top-down estimates and their interpretations (Sheng et al., 2019). Spatially, emissions hotspots (grid > 33 g CH4 m-2 yr-1) in PKU and EDGAR were generally located in the North China Plain and south China, which are densely populated areas, energy production regions, and agriculture-dominant regions. However, such patterns were not presented in GAINS and REAS, with a lack of emissions hotspots in the southern China and biased allocation of the majority emissions towards Shanxi provinces. The incomplete information on emission patterns may mislead or bias mitigation efforts for CH4 emission reductions. These findings provide useful information for research groups developing emission inventories; improve understanding of China's CH4 emissions; for targeting mitigation efforts; and reduce estimates uncertainty. Third, we have provided all the gathered datasets in a more publicly obtainable place (Table S2 and https://figshare.com/articles/dataset/Data_zip/12720989 (Lin et al., 2020) with a DOI (10.6084/m9.figshare.12720989.v2), many of them were previously documented in literatures but can hardly be available. Last but not least, this paper is a review article type in ESSD rather than a research article. Review and data synthesis is a common and traditional type of research. For example, Saunois et al. (2020) and Andrew (2020) reviewed the global CH4 emissions and sinks and global fossil fuel CO2 emissions, respectively. For China, Han et al. (2020) recently conducted a study on China's fossil fuel CO2 comparison. All of these studies collect as many data sets as possible on a topic and compared them in a systematical way: such as total time-series emissions, spatial patterns, and time changes in spatial distributions. The conclusion and implications of such studies can have significant importance on both scientific and social communities.

Second, the authors just compared the emission values between different inventories, however, such a simple analysis and comparison cannot provide us an evaluation of anthropogenic CH4 emissions inventories (mentioned by the paper title). Many explanations of the discrepancies between different inventories did not provide any evidence and cannot fully convince me. Response: This study synthesizes the publicly available emissions datasets, and then compared them in detailed source categories and explained the differences quantitatively where possible. Some discrepancies were because that their estimates are largely depend on national-based activity data and defaulted emission factors, which hardly fully interpret the variation of local condition, and characteristics of emission sources. However, some differences were difficult to explain without further input data (e.g. proxy data). Moreover, we revised the title to "A comparative study of anthropogenic CH4 emissions over China based on the ensemble of bottom-up inventories". Besides the comprehensive comparisons, we also provided composite estimates for both national total and sectoral emissions based on

these data sets (Fig. 5). Before the analyses in this paper, people even do not have a whole picture of the total and sectoral emissions and do not have a clear reference system. Moreover, we improved explanations by dig into the original data, e.g., in Lines 218-222, Lines 282-284 and Lines 333-335 to explore estimates differences in key emitters: rice cultivation, coal mines, and waste treatment.

Third, the analysis of emission spatial distributions did not make sense because the global and regional inventories listed in Table 1 typically allocated the country- and province-level emission estimates to grid cells to create emission maps. Some of the selected spatial allocation proxies are rather arbitrary in my opinion, which can-not provide us accurate emission mapping results. Therefore, the comparisons shown in Figures 2 to 4 cannot give us any useful information. Response: We admit that the spatial resolution of the datasets depends highly on the original country- or province-level emissions estimates. There can be one of the main reasons for the large uncertainties when allocating the large-scale data to small grids. The utilization of proxies, though arbitrary to some extent, is helpful for down-scaling the country/region specific emissions into grids when proxies present spatial details relvent to methane emissions. This is the acceptable approach in many existing studies that provided spatial allocation of emission estimates. The widely used EDGAR gridded dataset in atmospheric transport and inversion studies set a good example. Using energy sector (mainly controlled by coal mining) as an example (Fig. 2), PKU disaggregates the provincial activity data using the geolocations of coal mines from Liu et al. (2015) (4264 sites), and thus its spatial distribution is more reliable, which is further validated by Sheng et al. (2019) (Fig. 4l). EDGAR v4.2 originally used 328 coal mines with locations for China from world coal association (https://eerscmap.usgs.gov/wocqi) as point emissions to disaggregate the amount of national emissions (Greet et al., 2019), and then updates locations from Liu et al. (2015). However, emissions from coal mining estimated by EDGAR still have notable biases toward Shanxi province (Fig. 4f). Emissions produced by GAINS and REAS also show a clustered spatial distribution in the North China Plain (Fig. 2n and 2r). Indeed, spatial proxy data plays an important role in determining the distribution of

CH4 emissions, and regional activity data and localized emission factors also strongly influence the emissions pattern. Moreover, for rice cultivation emissions, PKU and Zhang et al., (2017) both used provincial cultivating areas and thus showed very consistent spatial distributions in southern China, while EDGAR used IRRI data and produced high emissions in Fujian and Zhejiang provinces, where rice areas are not so large. The objective of this comparative study is to analyze the differences among existing datasets when they applied diverse approaches to produce spatial and temporal detailed information of the methane emissions. We are not to defend the applicability of their approaches but focus on the magnitude and characteristics of the differences and the implication for datasets revision and policy-making. Although the spatial distribution of China's CH4 emissions was presented by previous individual studies, the comprehensive presentations are limited. Fig. 2 and 4 not only present the magnitude and spatial location of each contributing sources emissions, but also quantify the frequency of emissions at grid cell level to identify the features of spatial discrepancy among inventories (Fig. 2 q-t). For example, the emission frequency of energy sector revealed that EDGAR was largely determined by a large proportion of high emitting grids (grid cell> 60 g CH4 m-2 yr-1, 75% of energy emissions, Fig. 2r), which may lead to spatial bias in top-down estimates. Moreover, estimates from more detailed subsectors are helpful to explain the source of differences between inventories (for example the coal mining emissions). As illustrated in Fig. 4, REAS presents lower estimates in rice cultivation (9%-107%) and in livestock (2%-34%) than other inventories, while 50%-57% higher estimates in coal mining. Furthermore, independent detailed subsector data, e.g. Sheng et al. (2019) developed coal mine emissions based on more than 10000 coal mines, Lin et al. (2011) produced livestock emissions based on county-level activity data, and Zhang et al. (2017) simulated rice emissions by using the detailed regional water management data and provincial organic matter application rates, are used to evaluate the amount and spatial pattern of estimates among inventories (Fig. 4j-o).

Finally, I do not agree that the authors said "This study, to the best of our knowledge,

provides the first quantitative analysis of the amount and spatiotemporal patterns of CH4 emissions in China" in the conclusion section because of my comments above. Response: Thanks for the comment. After major revisions and the literature review we did, we think 'comprehensive' is more appropriate than the word 'best' in this conclusive sentence. We collected and formed a comprehensive datasets (13 inventories), including 5 gridded datasets (global and regional scale) and 8 published tabular datasets (national and provincial level) (Table 1 and Table S1), and have presented a comprehensive review of China's CH4 emissions. The magnitude, spatial distribution, inter-annual variability, and source contributions of emissions among inventories are compared in a systematical way to improve understanding China's CH4 emissions and its uncertainty. Our result show that anthropogenic CH4 emissions in China differ widely among inventories (16 Tg CH4 yr-1) in 2010 (Table S2), and reliable estimates of their differences and exploring the reasons are highly important. Our works shed light on the sources of differences and uncertainties among the current available inventories, and provide some suggestions for developing and optimizing CH4 emission estimates, especially for high CH4 emitting regions.

Overall, I don't see much scientific significance in this paper though it summarized plenty of data and did some analysis. The paper is not well written and needs lots of editing. I fear I cannot recommend this paper for publication in its current form Response: As an country with widespread rice and coal production areas and a growing human population with billions of people, China is a large emitter of CH4. A lot of studies have produced estimates of the methane emission from sources in China and datasets with spatial details have been compiled. Those datasets differed greatly not only in national/regional magnitudes but also in spatial patterns and temporal variations owing to many reasons including, as mentioned by the reviewer, the downscaling approaches with proxies. The scientific significance of this study is to make comprehensive comparison of the existing datasets of methane emissions from sources of China in order to figure out in what ways and to what extent they differed. The results of the comparative analysis is useful for the revision of the datasets and the further

studies and policy-making concerning methane emissions in China. In this revision of the MS, we have carefully addressed all the comments and suggestions of the two reviewers and have dug further into the datasets we collected. We also have invited native English editor to thoroughly polish our language. Here, we briefly summarize the highlights of our study: 1) Using 13 state-of-the-art inventories, we provide a comprehensive comparison and analyses on China's anthropogenic CH4 emissions, which would be done by individual studies and not yet conducted in previous studies. Detailed source categories are considered to identify the discrepancies and sources of uncertainty among inventories, which have a great implication for both researchers and policy makers; 2) We collected and provided key datasets for inventory development, the sector-specific emission factors and proxy data in Table S3-S6. We hope these major revisions and improvements will address the concerns of the reviewer.

Reference Andrew, R. M.: A comparison of estimates of global carbon dioxide emissions from fossil carbon sources, Earth System Science Data, 12, 1437-1465, 2020. Höglund-Isaksson, L.: Global anthropogenic methane emissions 2005-2030: technical mitigation potentials and costs, Atmospheric Chemistry and Physics, 12, 9079-9096, 2012. Han, P., Zeng, N., Oda, T., Lin, X., Crippa, M., Guan, D., Janssens-Maenhout, G., Ma, X., Liu, Z., and Shan, Y.: Evaluating China's fossil-fuel CO 2 emissions from a comprehensive dataset of nine inventories, Atmospheric Chemistry and Physics, 20, 11371-11385, 2020. Kurokawa, J. and Ohara, T.: Long-term historical trends in air pollutant emissions in Asia: Regional Emission inventory in ASia (REAS) version 3, Atmospheric Chemistry and Physics, 20, 12761-12793, 2020. Lin, Y., Zhang, W., and Huang, Y.: Estimating spatiotemporal dynamics of methane emissions from livestock in China, Environmental Science (in Chinese), 32, 2212-2220, 2011. Liu, Z., Guan, D., Wei, W., Davis, S. J., Ciais, P., Bai, J., Peng, S., Zhang, Q., Hubacek, K., and Marland, G.: Reduced carbon emission estimates from fossil fuel combustion and cement production in China, Nature, 524, 335-338, 2015. Saunois, M., Stavert, A. R., Poulter, B., Bousquet, P., Canadell, J. G., Jackson, R. B., Raymond, P. A., Dlugokencky, E. J., Houweling, S., and Patra, P. K.: The global methane budget 2000-2017, Earth

System Science Data, 2020. doi: 10.5194/essd-5112-1561-2020, 2020. Sheng, J., Song, S., Zhang, Y., Prinn, R. G., and Janssens-Maenhout, G.: Bottom-up estimates of coal mine methane emissions in China: a gridded inventory, emission factors, and trends, Environmental Science & Technology Letters, 6, 473-478, 2019. Zhang, W., Sun, W., and Li, T.: Uncertainties in the national inventory of methane emissions from rice cultivation: field measurements and modeling approaches, Biogeosciences, 14, 163-176, 2017.

Please also note the supplement to this comment:
https://essd.copernicus.org/preprints/essd-2020-210/essd-2020-210-AC2-supplement.pdf
* * *

---

## Editor Comment (EC1) · Nellie Elguindi (Editor) · 24 Dec 2020

Dear Xiaohui et al.,

Two referees have posted reviews of your manuscript requiring major revisions. Both reviewers have expressed concerns about the depth of the analysis as well as the amount of new information the paper brings to the community, particularly regarding sources of uncertainties in methane emissions in China. In submitting a revised manuscript, I encourage the authors to carefully consider whether they adequately address all of the concerns and comments detailed by the two reviewers.

Additionally, I'm concerned by the fact that the inventories presented in this study are outdated and don't provide any information on recent trends in methane emissions in

[Figure]

China which have most likely changed significantly. The authors should make every effort possible to include more recent estimates of methane emissions which would be of great interest to the community and necessary information for policy makers. If this information does not exist, or is not publicly available, please speculate as to why this is the case in the manuscript. For example, in the authors' response to Reviewer #1 it is stated that the REAS inventory no longer includes methane emissions. Please inquire or speculate as to why. If there really are no recent estimates of methane emissions for China, perhaps the authors could use proxies to infer, at least qualitatively, how emissions have changed in recent years. For example, many coal-fired power plants have closed recently following the implementation of strict air pollutant policies in China after 2012. If a revised manuscript is submitted, please include a thorough discussion addressing all these points regarding recent trends in the last 5-10 years. This discussion s viewed as very important to a paper whose goal is to provide a comprehensive evaluation of China's methane emissions to aide in climate change mitigation.

Please note that a revised manuscript does not necessarily guarantee acceptance and could be subject to additional reviews.
* * *

---

## Author Response (AR3)

Two referees have posted reviews of your manuscript requiring major revisions. Both Reviewers have expressed concerns about the depth of the analysis as well as the amount of new information the paper brings to the community, particularly regarding sources of uncertainties in methane emissions in China. In submitting a revised manuscript, I encourage the authors to carefully consider whether they adequately address all of the concerns and comments detailed by the two reviewers.

Additionally, I'm concerned by the fact that the inventories presented in this study are outdated and don't provide any information on recent trends in methane emissions in China which have most likely changed significantly. The authors should make every effort possible to include more recent estimates of methane emissions which would be of great interest to the community and necessary information for policy makers. If this information does not exist, or is not publicly available, please speculate as to why this is the case in the manuscript. For example, in the authors' response to Reviewer #1 it is stated that the REAS inventory no longer includes methane emissions. Please inquire or speculate as to why. If there really are no recent estimates of methane emissions for China, perhaps the authors could use proxies to infer, at least qualitatively, how emissions have changed in recent years. For example, many coal-fired power plants have closed recently following the implementation of strict air pollutant policies in China after 2012. If a revised manuscript is submitted, please include a thorough discussion addressing all these points regarding recent trends in the last 5-10 years. This discussion viewed as very important to a paper whose goal is to provide a comprehensive evaluation of China's methane emissions to aide in climate change mitigation.

Please note that a revised manuscript does not necessarily guarantee acceptance and could be subject to additional reviews.

Response: Thank you for your invaluable comments.

Firstly, we conducted an extensive review focusing on recent 5-10 years' literatures and included the important references (e.g., Miller et al., 2019; Sheng et al., 2020) in our results and discussions. It presented the most updated outcomes and enhanced our understanding of the recent trends of China's $CH_4$ emissions. China's coal mine emissions have become flatten since 2012, which dominate present-day $CH_4$ emissions trend. We added discussions in Lines 165-168 in the revised MS.

Secondly, we estimated China's $CH_4$ emissions using IPCC Tier 1 method based on the national activity data from the National Bureau of Statistics of China (NBS) and localized optimized emission factors that we compiled in this MS (Table S5-S7) during 2015-2019. Our estimates show that the $CH_4$ emissions in China increased slightly in recent years, which can be attributed to the slowdown of coal and agricultural emissions and slight increase in waste emissions. We estimated an increasing trend of 0.5 Tg $CH_4$ $yr^{-2}$ for the period of 2015-2019, which is rather consistent with the values (0.3 $\pm$0.1 Tg $CH_4$ $yr^{-2}$) estimated from the top-down approach by (Sheng et al., 2020). Furthermore, we improved the understanding of the recent coal prodcution in China according to the national activity data and published literature (Sheng et al., 2019; Sheng et al., 2020). We added these contents in results and discussions in Lines 161-165 and Lines 198-201 in the revised MS.

As for the REAS inventory, we communicatd with Dr. Jun-ichi Kurokawa. $CH_4$ is not included in the latest version of REAS (REASv3.2), because REASv3.2 was developed based on a domestic

fund where CH4 was out of scope. And we explained this reason in the response to Reviewer #1. Lastly, we thoroughly checked and revised the responses to the two reviewers and updated the above new findings.

Before taking a decision regarding your manuscript, I ask that you please address the editor's comments which were posted on December 24th. In particular, it's of concern that your comparison is for inventories which are not up-to-date, especially given the rapid changes that have occurred during the past decade in China due to the implementation of stringent policies.

I suggest that you make every effort to include more updated CH4 emissions in your comparison which would be of more value to the scientific community. For example, the CEDS global emissions inventory has been recently updated (https://github.com/JGCRI/CEDS/) to the year 2018. I suggest you contact the CEDS manager (Steve Smith) to ask if the CH4 emissions for China are available and could be included in your paper. In addition, you could consider including FAOSTAT emission estimates (http://www.fao.org/faostat/en/#data/EM) which go up to 2017. A discussion on the lack of recent data, and speculations as to why this is the case, would also be appropriate so that readers understand the limitations. Perhaps it would also be useful to contact the authors of the REAS inventory to enquire about why the CH4 emissions have not been updated.

Lastly, please make sure that your revised manuscript has be properly edited for grammatical errors and mispellings. I noticed several as I read through the revised manuscript.

Response: Thank you for your suggestions.

We communicated with Dr. Steve Smith and he kindly provided the recently updated $CH_4$ emissions data. We have included this new data of CEDSv2021-02-05 with the years from 2015-2019. We compared it with our independent recent estimates for 2015-2019 and found they are quite consistent in both total and sectoral emissions (especially for the energy sector), and also close to the national reported values to UNFCCC, which is encouraging.

The FAOSTAT emission estimates during 1990-2017 have been included in this study (Fig. 1).

For the lack of recent data, bottom-up estimates of $CH_4$ emissions are based on both activity data and emission factors. The sources of these data are mainly from national or provincial statistics including energy (coal, oil and natural gas), agriculture (rice areas and livestock numbers) and waste (landfills and waste water), and these data generally lags 1-3 years. Considering the time for inventory compiling, the $CH_4$ inventory would lag 2-4 years.

In the revised paper, the language has been polished by a native editor.

We hope these major improvements and revisions can address the concerns of Reviewers and Editor.

Refenerce

Sheng, J., Tunnicliffe, R., Ganesan, A., Maasakkers, J., Shen, L., Prinn, R., Song, S., Zhang, Y., Scarpelli, T., and Bloom, A.: Sustained methane emissions from China after 2012 despite declining coal production and rice-cultivated area, 2020. 2020.

Sheng, J., Song, S., Zhang, Y., Prinn, R. G., and Janssens-Maenhout, G.: Bottom-up estimates of coal mine methane emissions in China: a gridded inventory, emission factors, and trends, Environmental Science & Technology Letters, 6, 473-478, 2019.

Miller, S. M., Michalak, A. M., Detmers, R. G., Hasekamp, O. P., Bruhwiler, L. M., and Schwietzke, S.: China's coal mine methane regulations have not curbed growing emissions, Nature communications, 10, 1-8, 2019.